# ResLR: Residual-Low-Rank Surrogates for Stable and Fast Context Adaptive Computing in Large Language Models

## Abstract

Large Language Models (LLMs) achieve state-of-the-art results on diverse tasks, yet inference remains expensive because every token traverses the full Transformer stack. Recent context adaptive computing methods mitigate this cost by token-wise layer skipping, but their per-layer routing is volatile, leading to accuracy oscillations and an extended fine-tuning process. We trace this instability to two issues: (i) direct skips violate the model's functional hierarchy, and (ii) per-layer routing fails to exploit the similarity of activations between neighboring layers. We therefore propose a unified acceleration framework addressing both problems. First, we introduce the Residual-Low-Rank (ResLR) surrogate, a lightweight bypass that distills the residual transformation between consecutive layers into a low-rank operator within a compact subspace, thus synthesizing the effect of the skipped layers and preserving hierarchy. Second, we devise Block-Wise Multi-Path Routing, which clusters neighboring layers into blocks and issues a single routing decision per block, explicitly leveraging activation similarity to stabilize computation and reduce gating overhead. The method integrates into standard LoRA fine-tuning without extra stages. Across question answering, mathematical reasoning, and commonsense inference benchmarks, it reduces FLOPs by 48%–52% and yields ∼1.9× wall-time speed-ups while outperforming static and dynamic baselines. With feature probing suggests a ∼90% functional preservation, variance analysis shows 42.3% lower score standard deviation and 53.7% more stable routing than layer-skipping approaches, establishing ResLR and block-wise routing as a robust approach for practical, low-cost LLM inference.

## 1 Introduction

Large Language Models (LLMs) unlock unprecedented performance but incur prohibitive inference costs measured in floating-point operations (FLOPs). To address this, the community has pursued two main avenues: static compression and dynamic inference. Static methods such as quantization (Qin et al., 2023) and pruning (Ma et al., 2023) produce uniformly smaller models but still enforce a fixed computational budget per token, regardless of its token-level importance for next-token prediction. In contrast, dynamic inference (Stojkovic et al., 2025) leverages conditional computation to tailor the workload at runtime. This paradigm has progressed from coarse-grained early-exit

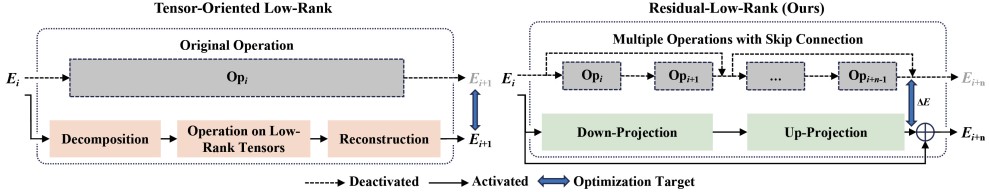

Figure 1: Conceptual overview of a Residual-Low-Rank (ResLR) surrogate. ResLR reformulates multiple residual blocks with skip connections into a unified architecture comprising a global skip connection and a low-rank residual, thereby approximating the transformation across a stack of Transformer layers efficiently.

strategies (Chen et al., 2024) to more fine-grained, token-level approaches that selectively execute Transformer layers for each input token. By allocating compute only where it is most useful, these methods offer a more precise and efficient response to the core challenges of LLM inference. Recent advances in dynamic computation (Jiang et al., 2024) further pioneer layer-skipping coupled with coordinated KV-cache eviction, demonstrating substantial potential to reduce inference-time costs.

Despite the promise of token-wise dynamic computation, our investigation reveals a critical limitation: routing decisions are fundamentally unstable. We attribute this instability to fluctuations in the effective depth experienced by each layer within the dynamic computation graph. While it is established that LLMs learn a functional hierarchy tied to layer depth during pre-training (Tenney et al., 2019), this concept is further operationalized in D-LLM by defining a layer's function as relative to specific semantic features. To this end, token-specialized skipping modules are designed to bypass layers with weak functional relevance (Jiang et al., 2024). However, the optimization process for this direct layer-skipping method does not fully leverage the pre-trained model's intrinsic functional hierarchy; instead, it learns a unstable hierarchical pattern. When depth varies across tokens, a layer cannot reliably infer a token's processing history from its representation, undermining the basis for coordinated, cross-layer decision-making. As a result, layer-wise gating degenerates into a greedy, localized strategy, with layers acting nearly independently. Empirically, we observe low mutual information between layer choices (Figure 2a) and elevated routing-variance (Figure 2b) across multiple validations on the SAMSum (Gliwa et al., 2019) benchmark; additional results on other benchmarks appear in Appendix E.2. This instability induces erratic optimization dynamics, prolonging fine-tuning relative to the static LoRA (Hu et al., 2022) baseline and increasing task-score variance versus static baselines. Consequently, our ResLR framework is driven by two objectives: (i) to introduce a lightweight computational surrogate that preserves the functionality of bypassed layers, and (ii) to restructure routing so as to exploit inter-layer correlation for improved routing decisions.

In this work, motivated by the low-rank nature of residuals between adjacent Transformer layers, we propose Residual-Low-Rank (ResLR) surrogates as a bypass structure (Figure 1). The framework explicitly learns transformation residuals via self-distillation and performs the transformation in a low-rank representation space, enabling it to emulate multiple consecutive operators with skip connections. This departs from conventional, tensor-oriented low-rank decompositions and helps preserve the model's hierarchical functional specialization under dynamic execution. To address routing instability, we further introduce a block-wise multi-path routing mechanism grounded in inter-layer activation similarity. By bundling consecutive layers into blocks and making a unified routing decision at the first layer of each block, this mechanism reduces routing overhead and improves the stability of both routing and overall model performance. The proposed acceleration framework integrates seamlessly into standard LoRA fine-tuning pipelines, supporting one-stage, synergistic optimization of model quality and inference efficiency.

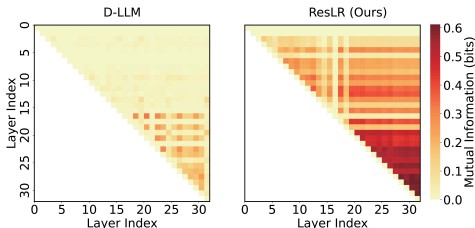

(a) Mutual information analysis of skipping decisions.

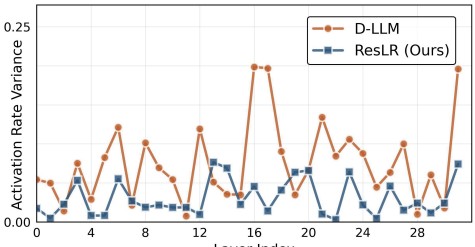

(b) Variance of layer activation rate.

Figure 2: To reveal routing decision instability in existing layer-skipping dynamic computing methods, we compare them against the representative layer-skipping D-LLM (Jiang et al., 2024) and the proposed method.

We evaluate ResLR across benchmarks for question answering, mathematical reasoning, and commonsense reasoning. ResLR reduces FLOPs by 48%–52% while outperforming both static fine-tuning (LoRA) baselines and state-of-the-art dynamic computation methods. These FLOPs savings translate to 1.9 wall-clock speedup with 94.6% scaling linearity. Stability improves as well, with a 42.3% reduction in task-score standard deviation and a 53.7% reduction in average routing-decision standard deviation relative to D-LLM Jiang et al. (2024). In summary, our contributions are:

- A self-distilled, low-rank surrogate that learns a functional operator to approximate multi-layer residuals, thereby preserving the model's core functional hierarchy.

- A block-wise multi-path routing scheme that exploits inter-layer similarity for stable, low-overhead gating.

- A one-stage fine-tuning framework that jointly optimizes accuracy and inference cost, achieving state-of-the-art FLOPs–performance trade-offs across multiple benchmarks.

## 2 RELATED WORK

### 2.1 DYNAMIC COMPUTATION OPTIMIZATION

The goal of dynamic computation is to selectively execute model components based on input features, thereby optimizing inference efficiency. Prevailing strategies include layer skipping, where decision modules determine whether to bypass entire layers (e.g., ConvNet-AIG (Veit & Belongie, 2018), SkipNet (Wang et al., 2018), Layerskip (Elhoushi et al., 2024)), and early exiting (Xu et al., 2025), which facilitates premature termination via intermediate classifiers when sufficient confidence is reached (e.g., MSDN (Chen et al., 2022)). These principles were successfully extended to the NLP domain, with models like DeeBERT (Xin et al., 2020) and FastBERT (Liu et al., 2020) adapting dynamic execution for Transformers. More recently, as models have scaled up, works like D-LLM (Jiang et al., 2024) have applied these ideas to large language models, allocating computational resources based on token importance. However, empirical observations suggest a fundamental trade-off exists between the fine-grained flexibility these methods provide and the stability of the model's training dynamics and final predictions.

### 2.2 LOW-RANK APPROXIMATION IN LLMS

Low-rank ideas permeate LLM optimization: most existing techniques simply apply a rank-$r$ decomposition to some tensors. When the tensor is a weight matrix, methods such as LoRA (Hu et al., 2022), AdaLoRA (Zhang et al., 2023), QLoRA (Dettmers et al., 2023), DoRA (Liu et al., 2024b), SVD-LLM (Wang et al., 2024) and TensorGPT (Xu et al., 2023) shrink the parameter space or storage cost but leave the forward computation unchanged. When the tensor lies in the activation pathway, approaches like Linformer (Wang et al., 2020), Nyströmformer (Xiong et al., 2021) and Multi-head Latent Attention (Liu et al., 2024a) project attention scores or values to a lower subspace, speeding up one sub-operation while still executing every Transformer block.

By contrast, we propose a residual-low-rank surrogate to reconstruct the inter-layer residual $\Delta E$ itself in a compact subspace at training time. At run time, a router then substitutes this surrogate for the full block, providing a procedural, switchable bypass rather than a static tensor factorization.

## 3 METHOD

To achieve efficient context-adaptive computation in large language models while mitigating routing instability, we propose a unified pipeline with two components. First, we introduce Residual-Low-Rank (ResLR) surrogates that learn via self-distillation and apply residual increments between adjacent layer embeddings within a learned low-rank subspace, thereby preserving the model's functional hierarchy under dynamic execution. Second, we design block-wise multi-path routing that groups neighboring layers into blocks and issues a single, activation-similarity–aware gating decision per block. These components are seamlessly integrated into a one-stage LoRA fine-tuning framework, as illustrated in Figure 3.

### 3.1 RESIDUAL-LOW-RANK MODELING

**Low-Rank Variations in Consecutive Layer Embeddings** In prior study (Liu et al., 2023), the authors highlight the presence of significant cosine similarity in embeddings between consecutive layers of LLMs. This proximity suggests that changes between these embeddings typically occur along limited dimensions. Such behavior motivates an examination of the rank characteristics of inter-layer embedding variations.

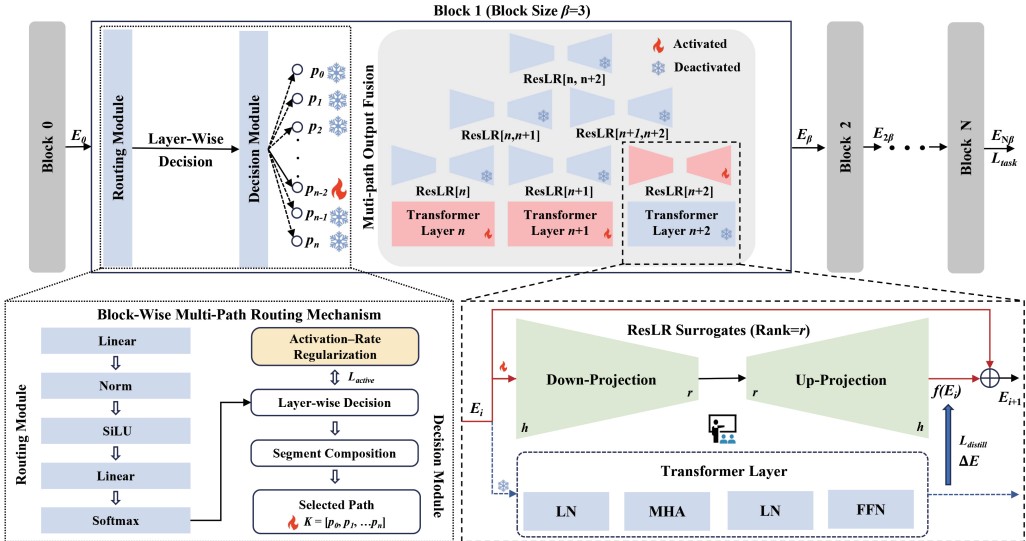

Figure 3: Overview of our unified pipeline for efficient, stable context-adaptive LLM inference. We propose ResLR surrogates to approximate residual increments between adjacent layer embeddings via low-rank projections to emulate multiple Transformer layers; and block-wise multi-path routing to issue a single, embedding-similarity-aware gate per layer block, all within a one-stage LoRA fine-tuning framework.

As illustrated in Figure 4a, we first validate the aforementioned inter-layer similarity on LLaMA2-7B fine-tuned with SAMSum dataset. Subsequently, we reformulate the variation behavior of the inter-layer embedding $E$ as the sum of the foundational embedding $E$ and the variation amount:

$$E_{i+1} = E_i + \Delta E. \tag{1}$$

By performing singular value decomposition on $\Delta E$ and ordering components by singular value magnitude, we observe pronounced low-rank structure. As shown in Figure 4b, the singular values drop sharply for indices below 50. To quantify this, we compute the information retention, defined as the ratio of the cumulative sum of squared singular values to the total sum of squared singular values. The results show that, for the LLaMA2-7B model with hidden dimension $H = 4096$, the first 50 components retain about $90\%$ of the information. This retention rises to $\sim 95\%$ when the number of components is increased to 256. Corresponding decompositions on other benchmarks are provided in Appendix E.1.

The observation of low-rank characteristics in inter-layer embedding variations reveals the existence of potential surrogate modules that can simulate the behavior of the corresponding original Transformer layers by modifying embeddings within a compressed feature space. This insight has inspired our framework design in subsequent sections.

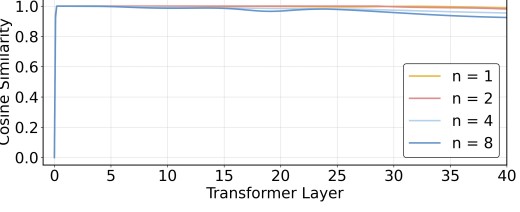

(a) Layer correlation on Transformer layers

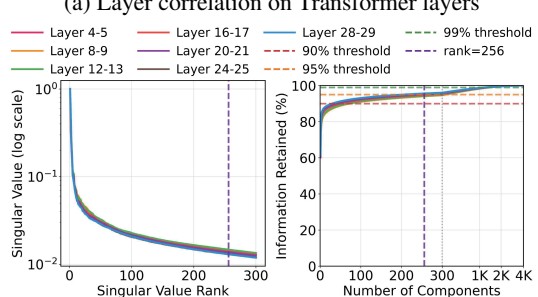

(b) Singular value decomposition and information retention

Figure 4: Verification of Low-Rank variations in consecutive Transformer layer embeddings.

**ResLR Surrogate for Transformation Retrieval**    In this section, we propose Residual-Low-Rank (ResLR) surrogate to replace the existing direct bypasses of Transformer layers, thereby facilitating efficient pathways for context-adaptive LLMs while ensuring improved functional preservation.

For a Transformer layer $i$ that includes layer normalization (LN), multi-head attention (MHA), and feedforward layers (FFN), the computational operations performed on the input embedding $E_i$ in this layer can be expressed as:

$$E_{i+1} = E_i + \text{MHA}(\text{LN}(E_i)) + \text{FFN}\left(\text{LN}\left(E_i + \text{MHA}(\text{LN}(E_i))\right)\right). \tag{2}$$

By comparing equation 1 and equation 2, we define the function of the surrogate module $f(\cdot)$ as follows:

$$f(E_i) \approx \text{MHA}(\text{LN}(E_i)) + \text{FFN}\left(\text{LN}\left(E_i + \text{MHA}(\text{LN}(E_i))\right)\right) = \Delta E. \tag{3}$$

Based on the known low-rank characteristics of $\Delta E$, $f(E_i)$ is allowed to perform transformation in the low-rank representation space. For simplicity, we use a dual-projection:

$$f(E_i) = W_u W_d E_i, \tag{4}$$

where $W_d \in \mathbb{R}^{d \times r}$, $W_u \in \mathbb{R}^{r \times d}$, and $r \ll d$ represents the low-rank dimension.

**Self-Distilling ResLR Surrogates**  Training a ResLR surrogate amounts to regressing the true residual $\Delta E$ with a rank-constrained map $f_\phi(E_i) = W_u W_d E_i$, whose parameters are $\phi = \{W_d, W_u\}$. Our aim is to minimise the mean-squared error:

$$\min_\phi \ \mathcal{L}_{\text{distill}}(\phi) = \mathbb{E}_{E_i}\left[\left\|\Delta E - f_\phi(E_i)\right\|_2^2\right]. \tag{5}$$

However, $\Delta E$ does not explicitly exist during the training process. Therefore, we create a skip connection for ResLR and treat the overall structure as a student model with the original transformer layer, computed in parallel, as the teacher, as shown in Figure 1. In practice, the computation formula for $\mathcal{L}_{\text{distill}}(\phi)$ is given by:

$$\min_\phi \ \mathcal{L}_{\text{distill}}(\phi) = \mathbb{E}_{E_i}\left[\left\|E_{\text{transformer}} - (E_i + f_\phi(E_i))\right\|_2^2\right]. \tag{6}$$

To analyze the error characteristics of $\mathcal{L}_{\text{distill}}$, we perform a bias-variance decomposition:

$$\Delta E - f_{\hat{\phi}}(E_i) = (\Delta E - f_{\phi^*}(E_i)) + (f_{\phi^*}(E_i) - f_{\hat{\phi}}(E_i)), \tag{7}$$

where $\hat{\phi}$ represents the learned parameters and $\phi^*$ denotes the optimal parameters, both defined within the rank-$r$ constrained hypothesis space $\mathcal{F}_r$.

Subsequently, we introduce Lemma 1 and defer its proof to Appendix A.

**Lemma 1:** Under the optimal parameters $\phi^*$, the error is orthogonal to the hypothesis space:

$$\mathbb{E}\left[\langle \Delta E - f_{\phi^*}(E_i), g(E_i)\rangle\right] = 0 \quad \forall g \in \mathcal{F}_r. \tag{8}$$

This indicates that the cross term for $\Delta E - f_{\phi^*}(E_i)$ and $f_{\phi^*} - f_{\hat{\phi}}$ within the expansion of the squared Euclidean distance is zero. Consequently, the population risk $\ell(\hat{\phi})$ can be rewritten as:

$$\ell(\hat{\phi}) = \mathbb{E}\left[\|\Delta E - f_{\phi^*}(E_i)\|_2^2\right] + \mathbb{E}\left[\|f_{\phi^*}(E_i) - f_{\hat{\phi}}(E_i)\|_2^2\right] = \epsilon_{\text{approx}} + \epsilon_{\text{est}}. \tag{9}$$

For the approximation component, stacking the true residuals of $n$ training tokens into $\mathbf{G} = [\Delta E^{(1)}, \ldots, \Delta E^{(n)}] \in \mathbb{R}^{d \times n}$ and denoting its singular values by $\sigma_1 \geq \cdots \geq \sigma_d$, the Eckart–Young–Mirsky theorem yields:

$$\epsilon_{\text{approx}} = \frac{1}{n}\left\|\mathbf{G} - \mathbf{G}_r\right\|_F^2 = \frac{1}{n}\sum_{j=r+1}^{d} \sigma_j^2, \tag{10}$$

thus indicating that the spectral tail after the $r$-th component fully determines the bias incurred by the rank constraint.

Meanwhile, the estimation error $\epsilon_{\text{est}}$ measures the gap between the learned surrogate $f_{\hat{\phi}}$ and the optimal rank-$r$ surrogate $f_{\phi^*}$. In our implementation, strong regularization via weight decay and

self-distillation suppresses $\epsilon_{\text{est}}$, and our rank selection experiment in Appendix D.1 shows that task performance shows improve-saturate characteristics with increasing $r$ without any degradation. Hence, under sufficient regularization, raising $r$ is effectively "safe": $\epsilon_{\text{approx}}$ remains dominant and no bias–variance reversal occurs.

Accordingly, we adopt a simplified rank-selection rule based on information retention. We choose the smallest rank $r$ that both satisfies our desired information threshold and meets the computational budget:

$$\frac{\sum_{j=1}^{r} \sigma_j^2}{\sum_{j=1}^{d} \sigma_j^2} \geq \eta_{\text{retention}} \quad \text{and} \quad C \leq C_{\text{budget}} \tag{11}$$

Here, the first term quantifies information retention as the cumulative fraction of total energy captured by the top $r$ components. We select $r$ to exceed a certain threshold $\eta_{\text{retention}}$ (e.g., $95\%$). The second term, $C$ bounds the additional computational overhead. Due to space limitations, a further discussion on the selection of $r$ and best practices is provided in Appendix D.1 and for the computational overhead analysis, see Appendix C.

## 3.2 BLOCK-WISE MULTI-PATH ROUTING MECHANISM

To further exploit the correlations of embeddings across successive layers and achieve more stable routing decisions, a Block-Wise Multi-Path Routing strategy is proposed, in which we organize consecutive $\beta$ layers into a block and generate unified routing decisions for the entire block.

**Router Architecture** Given the block-level input embedding $E_i \in \mathbb{R}^{T \times d}$, we instantiate a two–layer MLP router with SiLU and RMSNorm operators, as illustrated in Figure 3. During training, we utilize Gumbel–Softmax estimator to ensure each token decision $r_{t,j} \in \{0,1\}$ ($j = 0,\ldots,\beta-1$) is both binary valued and differentiable. At inference, the standard softmax is thresholded at $0.5$ to obtain deterministic gates.

**Path Composition and Output Fusion** At the network-initialization stage, we initialize a ResLR surrogate for every contiguous sub-segment $(s,e) \subseteq [0, \beta-1]$, denoted by $f_{s,e}(\cdot)$. After collecting the per-token binary gates $r_{t,j} \in \{0,1\}$, we merge consecutive layers whose gates are all zero into maximal low-rank segments $\mathcal{P} = \{(s_k, e_k)\}_{k=1}^{K}$ and invoke each $f_{s_k,e_k}$ exactly once. For token $t$ and any layer $j$, the hidden representation at the end of its enclosing segment is computed as

$$E_{t,e(j)+1} = r_{t,s(j)} \cdot \text{Transformer}_{s(j)}(E_{t,s(j)}) + (1 - r_{t,s(j)}) \cdot \left(E_{t,s(j)} + f_{s(j),e(j)}(E_{t,s(j)})\right), \tag{12}$$

where the interval selector functions $s(j)$ and $e(j)$ return, respectively, the start and end indices of the segment that contains layer $j$. If $r_{t,j} \neq 0$ (i.e., the full path is taken) or layer $j$ forms a singleton segment, then $s(j) = e(j) = j$; otherwise they identify the shared boundaries of a longer low-rank segment, indicating that the approximation $f_{s(j),e(j)}$ is executed only once for all layers within that segment. After traversing all $\beta$ layers, the block outputs $E_{i+\beta}$. We will further illustrate this process through an example in Appendix B.

**Activation–Rate Regularization** To explicitly regulate computational sparsity, we penalize the deviation between the empirical activation ratio $\alpha = \sum_{t>\gamma} \sum_j r_{t,j} / \left(\beta(T - \gamma)\right)$ and a target $\tau$. In the definition of $\alpha$, the first $\gamma$ are reserved out of a total of $T$ tokens, following the recommendation of Jiang et al. (2024). Accordingly, the activation–rate regularization term is:

$$\mathcal{L}_{\text{act}} = \left\| \alpha - \tau \right\|_1. \tag{13}$$

This term is optimized jointly with the task loss and self–distillation objective, guaranteeing stable convergence while meeting a pre-specified FLOPs budget.

In conclusion, the integrated ResLR and block-wise multi-path routing framework yields:

- **Compact multi-layer approximation:** ResLR captures the combined effect of several consecutive layers in a low-dimensional subspace, preserving end-to-end information flow.
- **Amortized routing cost:** A single block-level gating decision applies to all $\beta$ layers, leveraging inter-layer similarity to eliminate per-layer overhead and unstable decisions.

- **Reduced cumulative bias:** Consecutive low-rank segments are merged into one surrogate invocation, mitigating repeated approximation errors.

Finally, we introduce the overall loss function of ResLR. The objective function consists of three components: cross-entropy loss, activation loss, and distillation loss.

$$\mathcal{L}_{\text{total}} = \mathcal{L}_{\text{task}} + \lambda_{\text{active}} \cdot \mathcal{L}_{\text{active}} + \lambda_{\text{distill}} \cdot \mathcal{L}_{\text{distill}}, \tag{14}$$

where $\mathcal{L}_{\text{task}}$ is the cross-entropy loss for the main task, $\lambda_{\text{active}}$ is the hyperparameter for the activation loss $\mathcal{L}_{\text{active}}$, and $\lambda_{\text{distill}}$ is the hyperparameter for the self-distillation loss $\mathcal{L}_{\text{distill}}$.

## 4 EXPERIMENTS

### 4.1 EXPERIMENTAL SETUP

**Models and Benchmarks** Experiments are conducted using LLaMA2-7B, LLaMA2-13B, LLaMA3-8B and Qwen3-8B to validate the proposed ResLR method. We conducted a comparative analysis of four methods: MoD (Raposo et al., 2024), Shortened-LLaMA (Kim et al., 2024), Ada-Infer (Fan et al., 2024), and D-LLM. For these four approaches, we fine-tuned our model with self-distillation joint training on nine benchmarks, covering three categories of tasks. The first task is question answering and summarization, including Alpaca (Taori et al., 2023) and SAMSum (Gliwa et al., 2019). Second, GSM8K (Cobbe et al., 2021) and MaWPS (Kadlčík et al., 2023) are about math problem-solving. Answers are considered correct only when the numerical difference is less than $10^{-5}$. Finally, BoolQ (Clark et al., 2019), PIQA (Bisk et al., 2020), SIQA (Sap et al., 2019), OBQA (Mihaylov et al., 2018), and MMLU (Hendrycks et al., 2021) are datasets about common sense reasoning. Performance on these benchmarks is measured by accuracy.

**Implementation Details** All experiments were conducted on two NVIDIA A100 GPUs. Models were fine-tuned for four epochs (six epochs for Alpaca dataset) with a two-epoch linear warm-up. We fix the target activation rate to $\tau = 0.4$ and select the surrogate rank $r = 256$ and block size $\beta = 4$. The router hidden size is set to 512; we use $\lambda_{\text{distill}} = 1.0$ and set a unified activation-regularization weight $\lambda_{\text{active}} = 10.0$ for all tasks to balance the loss terms, as the task loss for most NLP benchmarks typically converges to a similar range. Detailed hyperparameter sensitivity analysis and best practices are presented in Appendix D. All models are trained with AdamW (with a 5e-4 learning rate and a 0.05 weight decay) and employ the KV-cache eviction strategy of D-LLM (Jiang et al., 2024) to reduce inference memory overhead. For LoRA parameters involved in fine-tuning, the rank is set fixed to 32, in alignment with our re-implementation for D-LLM.

### 4.2 PERFORMANCE COMPARISON

**Comparison with State-of-the-Art Methods** As shown in Table 1, we compared ResLR with state-of-the-art methods across nine benchmarks: question answering and summarization, mathematical reasoning, and commonsense reasoning. The baseline computational cost is normalized to 1.00 FLOPs for the full size model, with other methods reported as relative percentages. Taking into account all FLOPs overhead from the surrogates module (a detailed overhead analysis is provided in Appendix C), ResLR achieves superior performance across all nine benchmarks while consuming only $\sim 50\%$ of the computational cost. These results validate the advantages of ResLR over traditional layer skipping and single-layer routing methods, demonstrating a better balance between computational efficiency and model performance. We also conducted supplementary experiments on LLaMA2-13B to confirm that our method scales effectively to larger models. Additionally, for the representative datasets SAMSum, MaWPS, and BoolQ across the aforementioned three task categories, we have also plotted the FLOPs-performance Pareto Front to illustrate its superiority across all FLOPs budgets, as presented in Appendix E.4.

**Wall Time Speed-up Analysis** To verify the real-world efficiency of the proposed method, we conducted wall time measurements on LLaMA2-7B with a single A100 GPU. As shown in Table 2, our ResLR method achieves a significant wall time speed-up compared to baseline and current SOTA D-LLM (Jiang et al., 2024). ResLR achieves a $1.85\times$ speed-up over the baseline and a $1.25\times$ speed-up compared to D-LLM. These results indicate a correlation between

Table 1: Performance and computational cost comparison across various methods, with LLaMA2-7B as the baseline. Sh. Lla. PPL and Sh. Lla. Tay. refer to Shortened-LLaMA with the PPL and Taylor metrics (Jiang et al., 2024), respectively. Ada-Infer is not applicable to Q&A and Math tasks after introducing layer skipping. To validate scalability, we also report our method's performance on LLaMA2-13B, LLaMA3-8B and Qwen3-8B. Lower PPL (perplexity) and higher Acc (accuracy) indicate better performance; **bold** values represent superior performance coupled with computational savings.

| Method | Metrics | Q&A (PPL ↓) | | Math (Acc. ↑) | | Com. Sen. (Acc. ↑) | | | | |
|---|---|---|---|---|---|---|---|---|---|---|
| | | Alpaca | SAMSum | GSM8K | MaWPS | BoolQ | PIQA | SIQA | OBQA | MMLU |
| **Backbone** | | | | | **LLaMA2-7B** | | | | | |
| LoRA | Perf. | 4.69 | 3.61 | 0.27 | 0.72 | 0.73 | 0.84 | 0.81 | 0.79 | 0.54 |
| Finetune | FLOPs | 1.00 | 1.00 | 1.00 | 1.00 | 1.00 | 1.00 | 1.00 | 1.00 | 1.00 |
| MoD | Perf. | 10.32 | 4.47 | 0.08 | 0.33 | 0.64 | 0.49 | 0.58 | 0.42 | 0.28 |
| (2024) | FLOPs | 0.56 | 0.56 | 0.56 | 0.56 | 0.56 | 0.56 | 0.56 | 0.56 | 0.56 |
| Sh. Lla. PPL | Perf. | 7.09 | 4.39 | 0.10 | 0.52 | 0.67 | 0.76 | 0.75 | 0.63 | 0.47 |
| (2024) | FLOPs | 0.66 | 0.66 | 0.66 | 0.66 | 0.66 | 0.66 | 0.66 | 0.66 | 0.66 |
| Sh. Lla. Tay. | Perf. | 7.65 | 4.66 | 0.18 | 0.39 | 0.73 | 0.83 | 0.81 | 0.81 | 0.53 |
| (2024) | FLOPs | 0.66 | 0.66 | 0.66 | 0.66 | 0.66 | 0.66 | 0.66 | 0.66 | 0.66 |
| Ada-Infer | Perf. | 319 | 874 | 0.00 | 0.00 | 0.71 | 0.55 | 0.80 | 0.78 | 0.41 |
| (2024) | FLOPs | 0.65 | 0.56 | 0.83 | 0.90 | 0.61 | 0.63 | 0.64 | 0.76 | 0.60 |
| D-LLM | Perf. | 6.01 | 3.18 | 0.29 | 0.74 | 0.73 | 0.84 | 0.82 | 0.80 | 0.53 |
| (2024) | FLOPs | 0.59 | 0.55 | 0.59 | 0.56 | 0.52 | 0.52 | 0.54 | 0.53 | 0.55 |
| **ResLR** | Perf. | **2.97** | **2.65** | **0.32** | **0.81** | **0.85** | **0.84** | **0.82** | **0.81** | **0.54** |
| (Ours) | FLOPs | **0.52** | **0.51** | **0.50** | **0.49** | **0.48** | **0.47** | **0.47** | **0.48** | **0.49** |
| **Backbone** | | | | | **LLaMA2-13B** | | | | | |
| LoRA | Perf. | 4.25 | 3.08 | 0.35 | 0.87 | 0.81 | 0.84 | 0.81 | 0.82 | 0.55 |
| Finetune | FLOPs | 1.00 | 1.00 | 1.00 | 1.00 | 1.00 | 1.00 | 1.00 | 1.00 | 1.00 |
| **ResLR** | Perf. | **2.76** | **2.52** | **0.42** | **0.92** | **0.88** | **0.85** | **0.84** | **0.83** | **0.55** |
| (Ours) | FLOPs | **0.52** | **0.53** | **0.52** | **0.53** | **0.50** | **0.45** | **0.47** | **0.48** | **0.46** |
| **Backbone** | | | | | **LLaMA3-8B** | | | | | |
| LoRA | Perf. | 4.84 | 3.93 | 0.52 | 0.88 | 0.75 | 0.87 | 0.81 | 0.85 | 0.56 |
| Finetune | FLOPs | 1.00 | 1.00 | 1.00 | 1.00 | 1.00 | 1.00 | 1.00 | 1.00 | 1.00 |
| D-LLM | Perf. | 8.43 | 3.63 | 0.50 | 0.91 | 0.75 | 0.88 | 0.81 | 0.82 | 0.57 |
| (2024) | FLOPs | 0.59 | 0.58 | 0.60 | 0.55 | 0.51 | 0.52 | 0.55 | 0.53 | 0.55 |
| **ResLR** | Perf. | **3.02** | **2.85** | **0.53** | **0.93** | **0.84** | **0.88** | **0.84** | **0.86** | **0.58** |
| (Ours) | FLOPs | **0.51** | **0.52** | **0.54** | **0.52** | **0.48** | **0.47** | **0.51** | **0.50** | **0.51** |
| **Backbone** | | | | | **Qwen3-8B** | | | | | |
| LoRA | Perf. | 5.29 | 3.94 | 0.84 | 0.89 | 0.86 | 0.88 | 0.81 | 0.84 | 0.77 |
| Finetune | FLOPs | 1.00 | 1.00 | 1.00 | 1.00 | 1.00 | 1.00 | 1.00 | 1.00 | 1.00 |
| **ResLR** | Perf. | **3.96** | **3.51** | **0.85** | **0.93** | **0.88** | **0.89** | **0.83** | **0.86** | **0.78** |
| (Ours) | FLOPs | **0.51** | **0.55** | **0.56** | **0.53** | **0.49** | **0.48** | **0.50** | **0.51** | **0.50** |

FLOPs reduction and improvements in wall time. We employ the FLOPs reduction rate to the actual wall-time speed-up ratio as speed-up linearity; a value closer to 1 indicates lower overhead from auxiliary operations. This

Table 2: Wall time per token and relative FLOPs on LLaMA-2-7B (baseline normalized to 1.0). Linearity measures the ratio of FLOPs reduction to actual speed-up (Samsi et al., 2023); closer to 1 means lower overhead.

| Method | Wall time | FLOPs | Linearity |
|---|---|---|---|
| LLaMA2-7B | 38.400 ms | 1.00 | – |
| D-LLM (2024) | 25.856 ms | 0.59 | 87.7% |
| **ResLR(Ours)** | **20.704 ms** | **0.51** | **94.6%** |

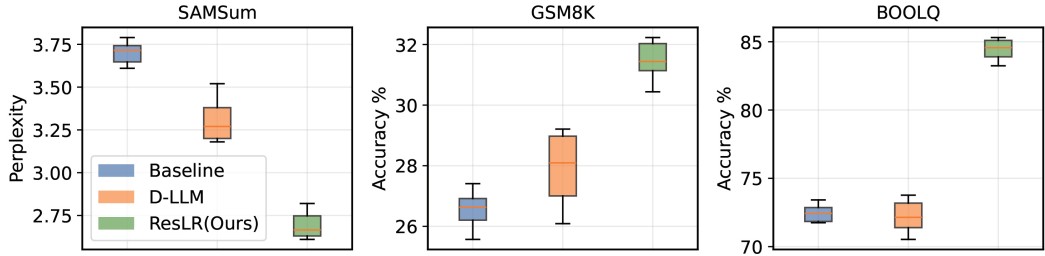

Figure 5: Standard deviation of results across different datasets, in comparison with LoRA (Hu et al., 2022) baseline and D-LLM.

higher linearity score indicates that our method reduces the computational cost while introducing less overhead. Quantitatively, ResLR introduces 2.75% more parameters and $< 1\%$ FLOPs overhead compared to the static baseline (see Appendix C for detailed overhead analysis). Moreover, since the routing decision is executed exclusively at the first layer of each block, the overhead associated with memory access caused by frequent path switching is further reduced, thereby enhancing the linearity of the observed speed-up.

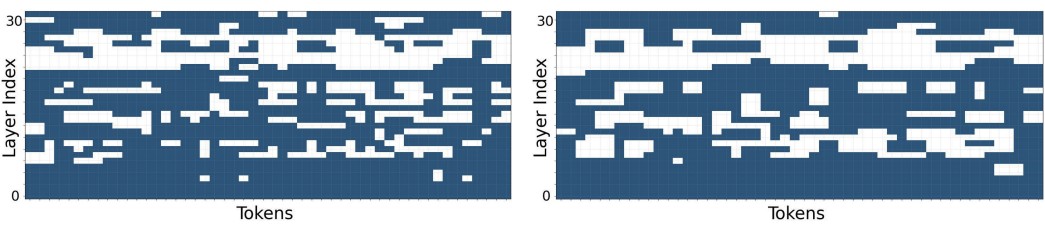

(a) Layer-wise bypass status for D-LLM

(b) Layer-wise bypass status for ResLR

Figure 6: Comparison of layer-wise bypass status for the same input token, where unfilled layers represent bypassed (skipped) layers, and vice versa.

**Routing Decision Stability** As shown in Figure 2, ResLR significantly improves decision stability, increasing inter-layer mutual information (Figure 2a) and reducing decision variance by 53.7% (Figure 2b). This stability, driven by ResLR surrogate and block-wise routing, directly translates to tangible benefits. In Figure 5, we show the results of 20 validation runs for the datasets SAMSum, GSM8K, and BoolQ. The box plots indicate that ResLR reduces the score standard deviation by 42.3% compared to D-LLM. Figure 7 illustrates that our framework reduces training instability, leading to a significantly shorter training cycle, thereby making

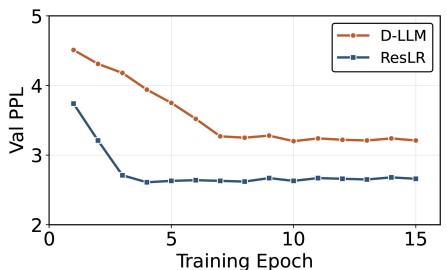

Figure 7: Training convergence curves of D-LLM and ResLR on SAMSum.

model adjustment and optimization more efficient. Additionally, visualizations of token activation states further confirm this structured behavior (Figure 6), revealing coherent decisions both across adjacent layers (y-axis) and within local semantic windows (x-axis), detailed visualization is presented in Appendix E.5. Due to space limitation, supplementary validation results on other task categories and benchmarks will be presented in Appendix E.2 and E.3.

**Illustration of Functional Hierarchy Preservation** To examine whether ResLR preserves depth-function mapping from original LLM, we analyze the mean activation rate of each Transformer layer across datasets (Fig. 8). For the direct-skipping baseline D-LLM, we observe a highly polarized pattern, and several Q&A benchmarks effectively collapse into a shallow subnetwork. This indicates that the direct skip-over mode discards the original model's functional hierarchy through optimization and establishes a new one. In contrast, ResLR exhibits a much smoother decay of activation with depth. In Appendix E.8, we conduct further probing experiments by calculating the point-to-point cosine similarity between the intermediate features of the original LLM and those of D-LLM

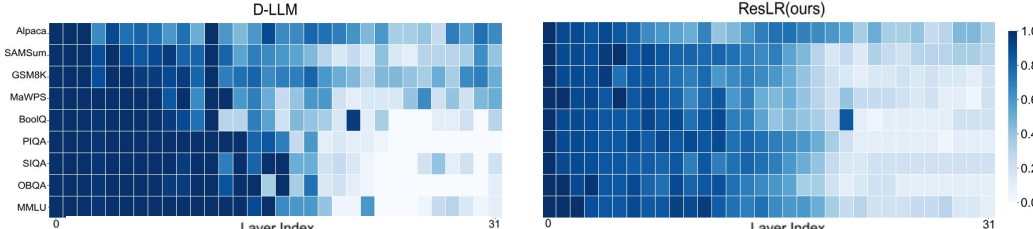

Figure 8: Mean layer activation rates across tasks for D-LLM and ResLR on LLaMA2-7B, the color intensity represents the activation rate, with darker shades indicating higher activation rates. D-LLM exhibits a highly polarized pattern and even collapses some tasks into shallow subnetworks, whereas ResLR maintains a smoother depth profile that keeps deeper layers engaged.

and ResLR, respectively. The results demonstrate that ResLR achieves approximately 90% layer functionality retention, which is a 26% improvement compared to D-LLM. Together with the lower routing variance in Fig. 2b, this suggests that our hierarchy-aware bypass behaves as a form of regularization: when a layer is not skipped, gradients still flow through the original Transformer block, encouraging each depth to retain a well-defined role instead of being replaced by a shallow shortcut. Consequently, ResLR outperforms both standard LoRA and all existing dynamic methods in task metrics at comparable or lower compute budgets in Table 1, supporting the view that preserving an effective depth profile is beneficial for downstream performance.

**Ablation Experiments** We conduct ablation experiments on LLaMA2-7B to evaluate the effectiveness of our proposed components. The experiments cover three representative tasks—question answering and summarization, mathematical problem solving, and commonsense reasoning. As shown in Table 3, integrating ResLR and the block router consistently improves performance over the baseline, with especially notable gains on mathematically complex tasks, underscoring ResLR's capacity to address reasoning-intensive scenarios.

## 5 CONCLUSION AND DISCUSSION

This study presents the ResLR framework, which effectively addresses the high computational costs of LLM inference and the performance instability inherent in dynamic, layer-skipping techniques. Our method successfully mitigates this issue while demonstrating significant acceleration. By introducing a lightweight surrogate structure and a block-wise multipath routing mechanism, we achieved a reduction of 48% to 52% in FLOPs across multiple benchmark tests, while also attaining up to $1.9\times$ speed-up in wall time without sacrificing performance. Additionally, with feature probing suggests a

Table 3: Ablation study isolating the contributions of our proposed components. We compare our full ResLR model against variants that remove the block router (w/o Block router) or use direct skip bypasses (w/o Surrogates), while D-LLM is used as a baseline without either component.

| Dataset | Alpaca | | GSM8K | | BoolQ | |
|---|---|---|---|---|---|---|
| Metrix | PPL | FLOPs | Acc. | FLOPs | Acc. | FLOPs |
| ResLR | **2.97** | **0.52** | **0.32** | **0.50** | **0.85** | **0.48** |
| w/o Block router | 3.35 | 0.54 | 0.30 | 0.54 | 0.84 | 0.51 |
| w/o Surrogates | 3.96 | 0.54 | 0.28 | 0.52 | 0.75 | 0.50 |
| D-LLM | 6.01 | 0.59 | 0.29 | 0.59 | 0.73 | 0.52 |

∼90% functional preservation, the standard deviation of task scores decreased by 42.3%, and the standard deviation of routing decisions decreased by 53.7%, indicating enhanced stability. Moreover, the functionality of the ResLR framework is defined by the decomposition of the transformer layer structure into its skip-connection and residual paths. This design is intended to preserve the functional hierarchy of the original model, thus exhibiting scalability and generalization ability. However, while the dual-projection ResLR surrogate effectively approximates most skipped operations, adding modest non-linear components could potentially further enhance its fidelity. Designing and analyzing such extensions would require more elaborate theoretical tools, as further discussed in Appendix E.6. Additionally, for batched input tokens, developing hardware-efficient computational paths for scattering inputs and gathering results remains a direction for future research.

## 6 REPRODUCIBILITY STATEMENT

To ensure reproducibility, we provide systematic pointers in the main text and appendices to implementation details, theoretical derivations, compute-overhead analysis, and supplementary experiments. Specifically: the method and training pipeline are summarized in Section 3, including the ResLR self-distillation objective 3.1 and Block-Wise Multi-Path Routing 3.2; implementation and training recipes (model and data, optimizer, number of epochs, target activation rate, rank and block size, etc.) are centralized in Section 4.1 to facilitate reproducing the experimental setup and environment; the theoretical part in Appendix A provides the key lemma and its complete proof to support methodological testability; a step-by-step example of the decide-once, execute-in-segments procedure for block-wise routing is given in Appendix B to reproduce the forward execution process; we present an itemized analysis and accounting of additional parameter and FLOPs overhead in Appendix C; hyperparameter sensitivity and best practices are summarized in Appendix D (including choices of rank $r$ and block size $\beta$, target activation rate and regularization coefficient recommendations) to guide resource budgeting and stable convergence; supplementary experiments (decision mutual information, variance stability, Pareto frontiers, and visualizations) are provided in Appendix E to cross-validate robustness and efficiency conclusions; the primary results, comparison baselines, and evaluation metrics are consolidated in Section 4.2 to enable verification under a consistent FLOPs budget and measurement. In addition, we provide anonymous downloadable demo code and run instructions in the **supplementary materials**.

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

# A    APPENDIX: PROOF OF LEMMA 1

In the main text, we explore the functional definition of the Residual-Low-Rank (ResLR) surrogate along with its optimization techniques. This section aims to substantiate the lemmas presented in the theoretical derivation by first reviewing the foundational framework that underpins ResLR modeling.

Consider a transformer layer $i$ comprising layer normalization (LN), multi-head attention (MHA), and feedforward layers (FFN). The computational operations performed on the input embedding $E_i$ in this layer can be articulated as follows:

$$E_{i+1} = E_i + \text{MHA}(\text{LN}(E_i)) + \text{FFN}\left(\text{LN}\left(E_i + \text{MHA}(\text{LN}(E_i))\right)\right). \quad (15)$$

We define the function of the surrogate module $f(\cdot)$ as approximately equal to:

$$f(E_i) \approx \text{MHA}(\text{LN}(E_i)) + \text{FFN}\left(\text{LN}\left(E_i + \text{MHA}(\text{LN}(E_i))\right)\right) = \Delta E. \quad (16)$$

To simplify the representation, we utilize a dual-projection approach:

$$f(E_i) = W_u W_d E_i, \tag{17}$$

where $W_d \in \mathbb{R}^{d \times r}$, $W_u \in \mathbb{R}^{r \times d}$, and $r \ll d$ denotes the low-rank dimension.

Training a ResLR surrogate requires regressing the true residual $\Delta E$ by employing a rank-constrained mapping $f_\phi(E_i) = W_u W_d E_i$, with the parameters specified as $\phi = \{W_d, W_u\}$. To achieve this, we minimize the mean-squared error defined as:

$$\min_\phi \ \mathcal{L}_{\text{distil}}(\phi) = \mathbb{E}_{E_i} \left[ \left\| \Delta E - f_\phi(E_i) \right\|_2^2 \right]. \tag{18}$$

To analyze the error characteristics associated with $\mathcal{L}_{\text{distil}}$, we conduct a bias-variance decomposition:

$$\Delta E - f_{\hat{\phi}}(E_i) = (\Delta E - f_{\phi^*}(E_i)) + (f_{\phi^*}(E_i) - f_{\hat{\phi}}(E_i)), \tag{19}$$

where $\hat{\phi}$ indicates the learned parameters, and $\phi^*$ represents the optimal parameters, both of which are defined within the rank-$r$ constrained hypothesis space $\mathcal{F}_r$.

Here, we present **Lemma 1:**

Under the optimal parameters $\phi^*$, the error vector $\Delta E - f_{\phi^*}(E_i)$ is orthogonal to the hypothesis space $\mathcal{F}_r$:

$$\mathbb{E}\left[\langle \Delta E - f_{\phi^*}(E_i), g(E_i) \rangle\right] = 0 \quad \forall g \in \mathcal{F}_r. \tag{20}$$

To prove this lemma, we consider the optimization problem on $\mathcal{F}_r$:

$$\phi^* = \arg \min_{\phi \in \mathcal{F}_r} J(\phi) \quad \text{where} \quad J(\phi) = \mathbb{E}\left[\|\Delta E - f_\phi(E_i)\|_2^2\right] \tag{21}$$

This objective function can be expanded as follows:

$$J(\phi) = \mathbb{E}\left[\langle \Delta E - f_\phi(E_i), \Delta E - f_\phi(E_i) \rangle\right] \tag{22}$$

At the optimal solution $\phi^*$, the Fréchet derivative of the objective function must be zero. For any direction $h \in \mathcal{F}_r$, we consider the perturbation $\phi^* + \epsilon h$:

$$\left. \frac{d}{d\epsilon} J(\phi^* + \epsilon h) \right|_{\epsilon=0} = 0 \tag{23}$$

Calculating the derivative yields:

$$
\begin{aligned}
\frac{d}{d\epsilon} J(\phi^* + \epsilon h) &= \mathbb{E}\left[\frac{d}{d\epsilon} \langle \Delta E - f_{\phi^* + \epsilon h}(E_i), \Delta E - f_{\phi^* + \epsilon h}(E_i) \rangle\right] \\
&= \mathbb{E}\left[-2 \left\langle \frac{df_{\phi^* + \epsilon h}(E_i)}{d\epsilon}, \Delta E - f_{\phi^* + \epsilon h}(E_i) \right\rangle\right] \\
&= \mathbb{E}\left[-2 \left\langle \frac{\partial f}{\partial \epsilon}, \Delta E - f_{\phi^* + \epsilon h}(E_i) \right\rangle\right]
\end{aligned} \tag{24}
$$

where $\frac{\partial f}{\partial \epsilon} = \lim_{\epsilon \to 0} \frac{f_{\phi^* + \epsilon h}(E_i) - f_{\phi^*}(E_i)}{\epsilon} = g_h(E_i)$ is the functional change corresponding to direction $h$, effectively representing the directional derivative.

At $\epsilon = 0$:

$$\left. \frac{d}{d\epsilon} J(\phi^* + \epsilon h) \right|_{\epsilon=0} = -2\mathbb{E}\left[\langle g_h(E_i), \Delta E - f_{\phi^*}(E_i) \rangle\right] = 0 \tag{25}$$

Therefore, we have:

$$\mathbb{E}\left[\langle g_h(E_i), \Delta E - f_{\phi^*}(E_i)\rangle\right] = 0 \quad \forall h \in \mathcal{F}_r \tag{26}$$

Since $h$ is an arbitrary direction and $\mathcal{F}_r$ is a linear space, $g_h$ can represent any function $g$ in $\mathcal{F}_r$:

$$\mathbb{E}\left[\langle \Delta E - f_{\phi^*}(E_i), g(E_i)\rangle\right] = 0 \quad \forall g \in \mathcal{F}_r \tag{27}$$

Therefore, we expand the squared norm in equation 18 as follows:

$$\begin{aligned}
\|\Delta E - f_{\hat{\phi}}(E_i)\|_2^2 &= \|(\Delta E - f_{\phi^*}(E_i)) + (f_{\phi^*}(E_i) - f_{\hat{\phi}}(E_i))\|_2^2 \\
&= \|\Delta E - f_{\phi^*}(E_i)\|_2^2 + \|f_{\phi^*}(E_i) - f_{\hat{\phi}}(E_i)\|_2^2 \\
&\quad + 2\langle \Delta E - f_{\phi^*}(E_i), f_{\phi^*}(E_i) - f_{\hat{\phi}}(E_i)\rangle
\end{aligned} \tag{28}$$

From the property in equation 27, we have that the third term satisfies:

$$\mathbb{E}\left[\langle \Delta E - f_{\phi^*}(E_i), f_{\phi^*}(E_i) - f_{\hat{\phi}}(E_i)\rangle\right] = 0 \tag{29}$$

As a result, we obtain the form presented in equation 8 of the main text.

# B APPENDIX: A CONCRETE EXAMPLE OF THE BLOCK-WISE MULTI-PATH ROUTING MECHANISM

To clarify the Block-Wise Multi-Path Routing mechanism described in Section 3 of the main text, this section provides a step-by-step walkthrough of a single forward pass for a token through a block. The core idea of this mechanism is to make a unified routing decision for a group of consecutive Transformer layers, thereby amortizing routing overhead and stabilizing the computational path.

Aligned with the experimental setup in our paper, a block is defined to contain $\beta = 4$ consecutive Transformer layers. We assume the current block starts at layer $i$, thus comprising layers $i, i+1, i+2$, and $i+3$. For a specific token $t$ in the sequence, its input embedding to this block is denoted as $E_{t,i}$. At the network initialization stage, the system pre-initializes a corresponding Residual-Low-Rank (ResLR) surrogate $f_{s,e}(\cdot)$ for every possible contiguous sub-segment $(s,e)$ within the block. For instance, $f_{0,0}$ acts as a surrogate for the single layer $i$, $f_{1,2}$ for the transformation across layers $i+1$ to $i+2$, and $f_{0,3}$ for the entire four-layer block.

## B.1 ROUTING DECISION

When the token embedding $E_{t,i}$ reaches the block's entrance (layer $i$), it is first fed into the block's router, which is a two-layer MLP. The router generates a binary gating vector $r_t$ of size $\beta$ for this token $t$. Let's assume the computed decision vector is: $r_t = [1, 0, 0, 1]$.

Each element in this vector corresponds to a layer within the block. A value of 1 indicates that the full Transformer computation for that layer should be executed (the "activation" path), while a "0" signifies that the layer should be bypassed using its ResLR surrogate (the "bypass" path). Thus, $r_t = [1, 0, 0, 1]$ implies:

- Layer $i$: Activated (execute Transformer$_i$)
- Layer $i + 1$: Bypassed
- Layer $i + 2$: Bypassed
- Layer $i + 3$: Activated (execute Transformer$_{i+3}$)

## B.2 PATH COMPOSITION AND SEGMENTATION

Based on the gating vector $r_t$, the system merges consecutive bypassed layers into "maximal low-rank segments": Layer $i$ has a gate value of 1 and forms an independent "activation segment". Layers $i+1$ and $i+2$ both have gate values of 0 and are contiguous, so they are merged into a single "low-rank segment". Layer $i+3$ has a gate value of 1 and also forms an independent "activation segment". Consequently, the computational path for token $t$ through this block is dynamically partitioned into three parts.

## B.3 COMPUTATION ALONG SEGMENTS

The token's embedding then proceeds through these three segments sequentially, strictly following Equation 12 from the main text:

**Process Segment 1 (Layer $i$):**

- Input: $E_{t,i}$
- Output: $E_{t,i+1} = \text{Transformer}_i(E_{t,i})$

**Process Segment 2 (Layers $i+1$ to $i+2$):**

- Input: $E_{t,i+1}$
- Output: $E_{t,i+3} = E_{t,i+1} + f_{1,2}(E_{t,i+1})$

The surrogate $f_{1,2}(\cdot)$ is invoked only once. Its output directly represents the state after traversing both layers. This prevents the accumulation of approximation errors that could arise from two separate, single-layer surrogate computations.

**Process Segment 3 (Layer $i+3$):**

- Input: $E_{t,i+3}$
- Output: $E_{t,i+4} = \text{Transformer}_i(E_{t,i+3})$

This example clearly illustrates that for any token entering a block, the system requires only a single router invocation to determine its complete computational path across all $\beta = 4$ layers. The routing decision $[1, 0, 0, 1]$ is ultimately translated into 2 full Transformer layer computations and 1 ResLR surrogate computation.

This "decide once, execute in segments" model not only minimizes routing overhead through amortization but also reduces cumulative approximation error by merging consecutive bypass layers and replacing multiple single-layer surrogates with a single, more accurate multi-layer surrogate $f_{1,2}$. Ultimately, this achieves a more stable and efficient context-adaptive computation.

## C  APPENDIX: COMPUTATIONAL OVERHEAD ANALYSIS

In the section on computational overhead analysis, we provide a detailed analysis of the parameter overhead and FLOPs introduced by ResLR, as mentioned in the main text. We utilize the model configuration adopted in the article. For the newly added modules, the ResLR rank is set to $r = 256$, the block size $\beta = 4$ with 8 blocks in total, and the dimensions of the block router hidden layer is $d_{\text{rout}} = 512$. The baseline model employs the standard Llama-2-7B (approximately 6.74 billion parameters) with an embedding dimension of $d_{\text{embedding}} = 4096$, a layer count of $n_{\text{layers}} = 32$, a number of heads $n_{\text{heads}} = 32$, and a head dimension of $d_{\text{head}} = 64$.

### C.1 PARAMETER OVERHEAD

For each ResLR surrogate, $2dr$ parameters are introduced. Within a single block, considering all sub-segments, there are $\beta(\beta + 1)/2$ surrogates. With 8 blocks, the total number of parameters is calculated as $2dr \times 10 \times 8 = 167,772,160$.

Furthermore, for the block router, a single module introduces:

- Linear1: $4096 \times 512 = 2,097,152$
- RMSNorm: $512$
- Linear2: $512 \times 8 = 4,096$

In total, this results in $2,101,760$ parameters. For 8 block routers, this adds an additional $16,814,080$ parameters. Thus, the proposed framework introduces approximately $184.6$ million parameters, accounting for $2.75\%$ of the Llama-2-7B model.

## C.2 FLOPS OVERHEAD

For the FLOPs overhead, each ResLR surrogate incurs the following FLOPs when processing a token:

- Up-Projection: $4096 \times 256 \times 2 = 2,097,152$
- Down-Projection: $256 \times 4096 \times 2 = 2,097,152$

The total is approximately 4.2M FLOPs. Additionally, the FLOPs for a single Transformer layer of Llama-2-7B is approximately 405M (considering only matrix multiplications). Therefore, the overhead introduced by a single surrogate accounts for $1.03\%$. Since each ResLR surrogate may be used to skip multiple Transformer layers, this comparison can be considered as the maximum ratio of the ResLR surrogate overhead.

Similarly, the FLOPs introduced by the block router primarily arise from Linear 1, with the specific value being $4096 \times 512 \times 2 \approx 4.19$ M.

## D APPENDIX: HYPERPARAMETER SENSITIVITY ANALYSIS AND BEST PRACTICES

In this section, we empirically analyze the hyperparameters discussed in the main text to demonstrate the practical hyperparameter tuning strategies of ResLR.

### D.1 RANK NUMBER AND BLOCK SIZE

| Metrix | r = 128 | | r = 256 | | r = 512 | |
|---|---|---|---|---|---|---|
| | PPL | FLOPs | PPL | FLOPs | PPL | FLOPs |
| $\beta = 2$ | 2.66 | 0.52 | 2.63 | 0.54 | 2.61 | 0.55 |
| $\beta = 4$ | 2.66 | 0.51 | 2.61 | 0.51 | 2.62 | 0.53 |
| $\beta = 8$ | 2.75 | 0.51 | 2.69 | 0.51 | 2.67 | 0.52 |
| $\beta = 16$ | 2.97 | 0.52 | 2.89 | 0.53 | 2.88 | 0.53 |

Table 4: Experiments on the optimal combination of the surrogate rank r and block size $\beta$ on the SAMSum dataset.

To validate the impact of different surrogate ranks $r$ and block sizes $\beta$ on model performance, we perform grid search on SAMSum benchmark with $r = \{128, 256, 512\}$ and $\beta = \{2, 4, 8, 16\}$ to demonstrate how the model performance typically changes with respect to these two hyperparameters. The results, as shown in Table 4, indicate that the model already achieves satisfactory task performance when $r = 256$, which is consistent with the $95\%$ information retention point observed in the main text. Further increasing $r$ leads to a saturation effect, resulting in diminished marginal gains. Regarding block size $\beta$, it is generally found that $\beta = 4$ is the optimal setting. Increasing $\beta$ further causes the number of possible paths within a single block to grow quadratically, which in turn decreases the frequency with which each surrogate is activated (and thus optimized) during backpropagation, ultimately increasing the training difficulty. On the other hand, setting $\beta$ too low reduces the utilization of inter-layer feature similarity and causes the accumulation of

greater approximation errors as well as decision latency, thereby resulting in a suboptimal performance–FLOPs trade-off.

Furthermore, in the main text, we discussed the relationship between $r$ and approximation error through a bias–variance decomposition, and noted that larger $r$ values can increase the likelihood of estimation error. To investigate this further, we plotted the rank–performance curve on the MaWPS dataset by sweeping $r$ from 32 to 1024. Since the MaWPS dataset contains a relatively small number of training samples ($\sim$240K tokens), it is particularly suitable for characterizing estimation error. As shown in Figure 6, due to the strong regularization effects of self-distillation and weight decay, even when $r = 1024$—where the number of parameters in a single surrogate is roughly 35 times the number of training samples—no significant performance degradation was observed. Instead, the overall task performance improves and then saturates as $r$ increases, demonstrating that increasing $r$ within a wide range is a safe strategy.

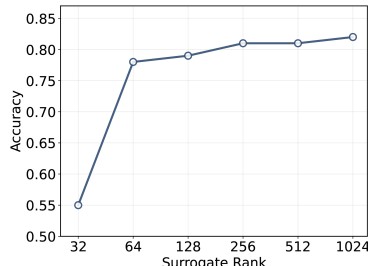

Figure 9: Performance Comparison of Different Surrogate Ranks of ResLR on the MaWPS Dataset.

Therefore, in practice (Figure 9), we recommend using $r = 256$ as a good start. After a single round of fine-tuning, the singular value decomposition of the inter-layer $\Delta E$ can be used as a reference to adjust $r$. For instance, if the 95% information retention point is above 256, $r$ can be increased accordingly. At the same time, the overhead calculation method described in Appendix C should be used to determine the computational resource budget.

### D.2 ACTIVATION–RATE REGULARIZATION

The activation–rate regularization term introduces two additional hyperparameters: the target activation rate $\tau$ and the loss coefficient $\lambda_{\text{active}}$. In the joint optimization problem of the proposed framework, the final achieved activation rate $\alpha$ tends to be slightly higher than $\tau$, enabling a trade-off among multiple optimization objectives. Therefore, $\tau$ should be set slightly lower than the desired value. For example, in our experimental setup described in the main text, we set $\tau = 0.4$ to achieve a final activation rate of approximately 0.5.

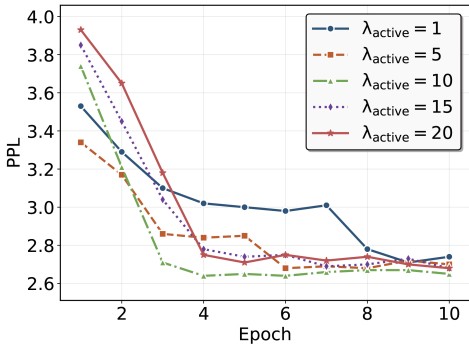

Figure 10: The effect of different $\lambda_{\text{active}}$ values on performance on the SAMSum dataset.

For $\lambda_{\text{active}}$, we performed a sweep from $\lambda_{\text{active}} = 1$ to $\lambda_{\text{active}} = 20$ on the SAMSum benchmark to investigate how its value affects the optimization dynamics. As shown in Figure 10, different values of $\lambda_{\text{active}}$ mainly impact the convergence speed of task performance; overall, the convergence speed initially increases and then decreases as $\lambda_{\text{active}}$ increases. A weaker regularization strength significantly prolongs the optimization cycle, whereas an excessively large $\lambda_{\text{active}}$ causes this term to become dominant in the early stages of training, thereby overshadowing the optimization directions of other terms.

In practice, the optimal value of $\lambda_{\text{active}}$ typically falls within the range of [1, 10]. Therefore, researchers may start with $\lambda_{\text{active}} = 10$ and observe the optimization curve of task performance, ensuring that the model exhibits noticeable convergence within the first 5 epochs. A well-chosen $\lambda_{\text{active}}$ usually leads to superior performance already in the early fine-tuning process.

## E APPENDIX: SUPPLEMENTARY EXPERIMENTAL RESULTS

In this section, we present additional experimental results mentioned in the main text.

### E.1 VALIDATION OF THE LOW-RANK RESIDUALS ON ADDITIONAL TASKS AND BENCHMARKS

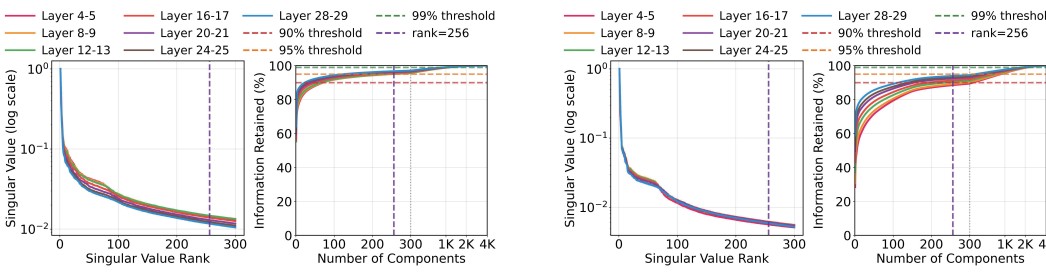

(a) Results on GSM8K benchmark     (b) Results on BoolQ benchmark

Figure 11: Singular values and information retention versus rank number for (a) GSM8K benchmark and (b) BoolQ benchmark.

To further validate the generality of the low-rank property of inter-layer residuals during fine-tuning across various tasks, we conducted additional residual SVD decomposition and information retention experiments on the GSM8K and BoolQ benchmarks, which correspond to mathematical reasoning and question answering tasks, respectively. As shown in Figure 11, for both benchmarks, the singular values obtained from SVD decomposition exhibit rapid decay in the early stages. Moreover, when the rank is set to 256, the information retention of most inter-layer residuals falls within the range of 90–95%. These results further substantiate the plausibility of employing low-rank approximation for inter-layer residuals.

### E.2 VALIDATION OF IMPROVED ROUTING DECISION MUTUAL INFORMATION ON ADDITIONAL TASKS AND BENCHMARKS

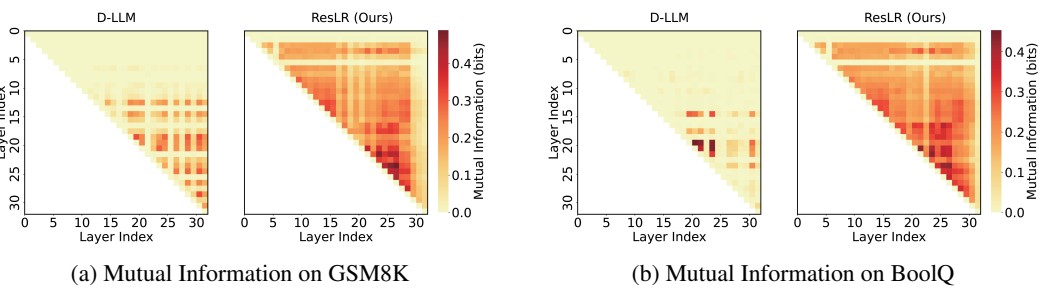

(a) Mutual Information on GSM8K     (b) Mutual Information on BoolQ

Figure 12: Mutual information experiments on additional datasets.

In Figure 12, we also conducted additional analyses on the layer-wise mutual information characteristics of ResLR decisions on the aforementioned two benchmarks. Compared to D-LLM, the higher inter-layer decision mutual information observed with our proposed method demonstrates that maintaining the model's functional hierarchy contributes to improved decision stability.

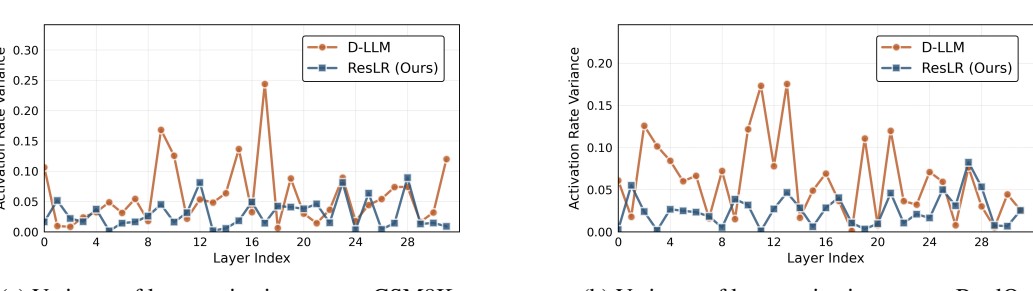

(a) Variance of layer activation rate on GSM8K     (b) Variance of layer activation rate on BoolQ

Figure 13: We analyze the instability of routing decisions in layer-skipping dynamic computation methods by calculating the variance of layer activation rates on GSM8K and BoolQ.

### E.3 VALIDATION OF IMPROVED OPTIMIZATION DYNAMICS ON ADDITIONAL TASKS AND BENCHMARKS

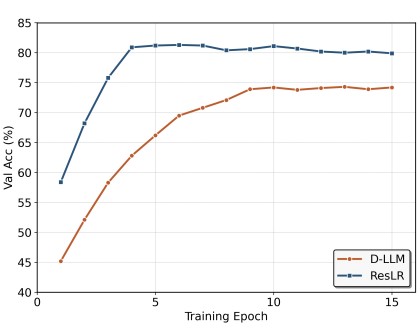 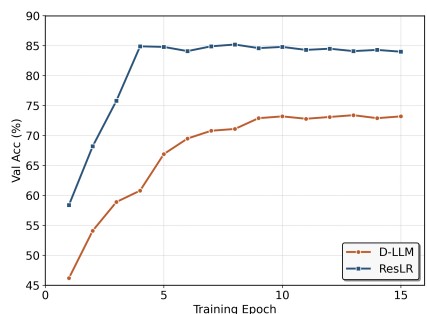

(a) MaWPS training performance                    (b) BoolQ training performance

Figure 14: Validate the self-distillation training characteristics of ResLR on the SAMSum, MaWPS, and BoolQ datasets. The x-axis represents epochs, and the y-axis represents the performance metrics on the validation set for each dataset.

In addition to Figure 7 in the main text, we further present the training curves of ResLR on the MaWPS and BoolQ benchmarks to illustrate the superior training dynamics offered by the proposed method. As shown in Figure 13, we analyze the variance of layer activation for ResLR and layer-skipping dynamic computation methods. As shown in Figure 14, D-LLM typically requires more than 10 fine-tuning epochs to achieve optimal performance, whereas ResLR usually needs only 4 epochs. This demonstrates a reduction in fine-tuning cycles by over 60%, further confirming that the stable training dynamics brought by ResLR are broadly applicable.

### E.4 FLOPS-PERFORMANCE PARETO FRONTIERS ON VARIOUS DOWNSTREAM TASKS

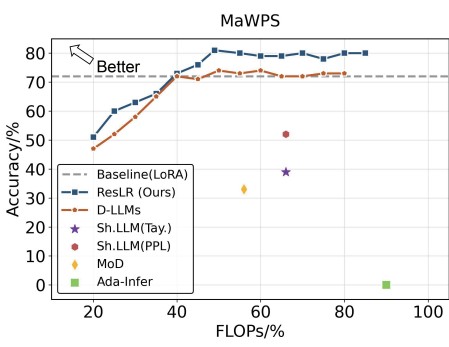 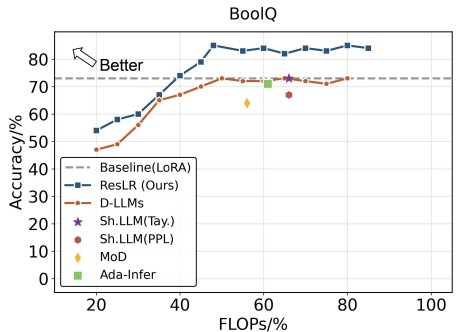

(a) Acc. and FLOPs performance on MaWPS         (b) Acc. and FLOPs performance on BoolQ

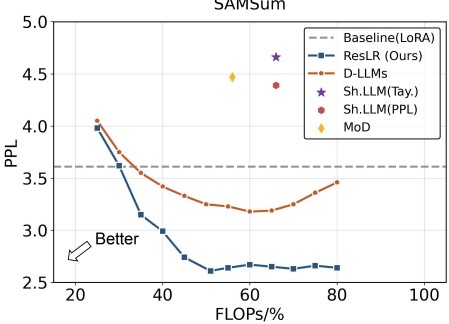

(c) PPL and FLOPs performance on SAMSum

Figure 15: Performance of ResLR on the SAMSum, MaWPS, and BoolQ datasets at different FLOPs.

We plot the FLOPs-performance Pareto frontiers on the downstream tasks discussed in the main text by varying the target activation rate $\tau$ across multiple repeated experiments. As shown in Figure 15, we present the results on SAMSum, MaWPS, and BoolQ. Compared to D-LLM, our method achieves superior performance across all FLOPs budgets. Moreover, ResLR effectively mitigates the overfitting observed with D-LLM at higher FLOPs budgets on the SAMSum dataset. Notably, performance on almost all tasks saturates at around 50% of the FLOPs budget, which may indicate an inherent sparsity property of the model.

### E.5 DETAILED VISUALIZATION OF LAYER-WISE BYPASS STATUS

In this section, we provide detailed information on the bypass activation state visualizations shown in Figure 6 of the main text. As shown in Figure 16, for both models, the input tokens and layer indices are fully aligned; the token segments are excerpted from the input prompt of the same sample in the SAMSum benchmark test set to ensure identical tokens across models.

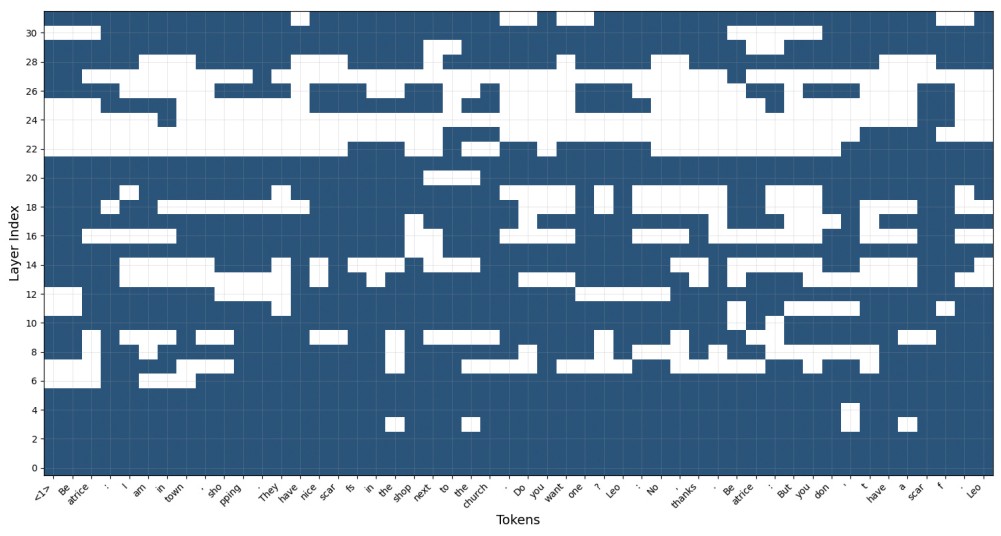

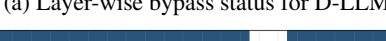

(a) Layer-wise bypass status for D-LLM

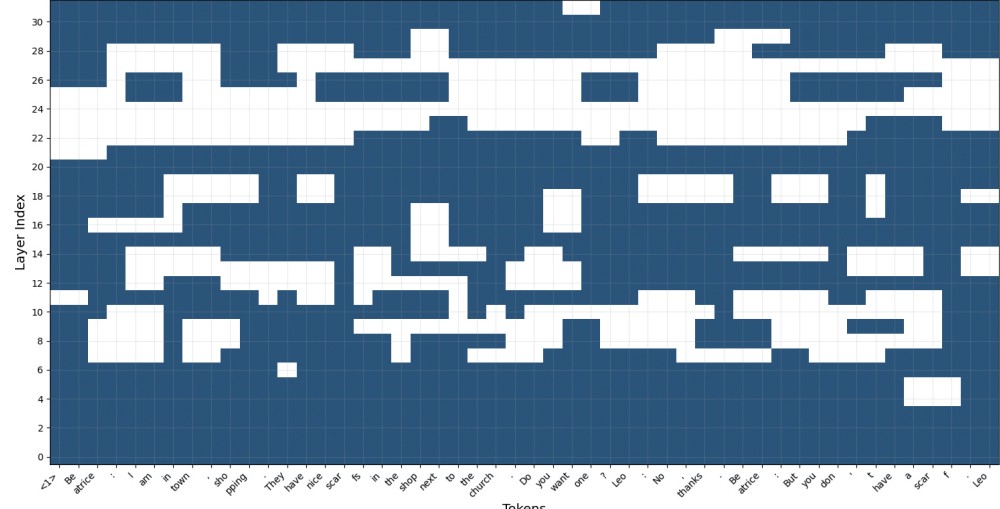

(b) Layer-wise bypass status for ResLR

Figure 16: Comparison of layer-wise bypass status for the same input token with complete display of the corresponding token content, where unfilled layers represent bypassed (skipped) layers, and vice versa.

## E.6 INTRODUCING NONLINEAR FACTORS INTO THE RESLR SURROGATE

As shown in Table 5, we introduce nonlinearity into ResLR by adding a SiLU activation layer between its two projection layers. The ablation experiments are conducted on SAMSum, GSM8K, and BoolQ. The results indicate that introducing SiLU only improves task performance on the SAMSum benchmark. However, after incorporating a nonlinear operator into the surrogate, for the nonlinear hypothesis space $\mathcal{F}_r$, it can no longer be guaranteed that $f_{\phi^*}(E_i) - f_{\hat{\phi}}(E_i) \in \mathcal{F}_r$, and thus equation 29 no longer holds. As a result, modeling the self-distillation loss characteristics of ResLR would require more assumptions and advanced analytical methods, suggesting a potential direction for future research.

|            | SAMSum |       | GSM8K |       | BoolQ |       |
| ---------- | ------ | ----- | ----- | ----- | ----- | ----- |
| Metrix     | PPL    | FLOPs | Acc.  | FLOPs | Acc.  | FLOPs |
| with SilU  | 2.59   | 0.51  | 0.31  | 0.50  | 0.85  | 0.52  |
| w/o. SilU  | 2.61   | 0.51  | 0.32  | 0.50  | 0.85  | 0.50  |

Table 5: Investigate the impact of introducing SilU in ResLR on the experimental results.

## E.7 APPENDIX: VISUALIZATION OF TYPICAL TASK OUTPUT

---

**Instruction:** James decides to run 3 sprints 3 times a week. He runs 60 meters each sprint. How many total meters does he run a week?

**LLaMA2-7B with LORA:(1.00 FLOPs)**
James runs 3 sprints each time. He does this 3 times per week. Each sprint is 60 meters.
So in one session: 3 * 60 = 180 meters
James does this 3 times per week.
Each session = 180 meters
Total per week = 180 * 3 = 540 meters
So James runs a total of 540 meters per week.

**Ours on LLaMA2-7B:(0.50 FLOPs)**
To find the total meters James runs in a week, we can calculate it as follows:
1. Calculate the total meters per sprint: 60 meters.
2. Calculate the total meters per week: 3 sprints * 3 times a week * 60 meters = 540 meters.
Thus, the final answer is: 540

---

**Instruction:** Toulouse has twice as many sheep as Charleston. Charleston has 4 times as many sheep as Seattle. How many sheep do Toulouse, Charleston, and Seattle have together if Seattle has 20 sheep?

**LLaMA2-7B with LORA:(1.00 FLOPs)**
Seattle has 20 sheep.
Charleston has 4 times as many sheep as Seattle.
Toulouse has twice as many sheep as Charleston.
Charleston has 4*20 = 80 sheep.
Toulouse has 80*2 = 160 sheep.
Altogether, Toulouse, Charleston, and Seattle have 80+160+20 =260 sheep.

**Ours on LLaMA2-7B:(0.50 FLOPs)**
Charleston has 4*20 = 80 sheep.
Toulouse has 80*2 = 160 sheep.
Altogether, Toulouse, Charleston, and Seattle have 80+160+20 = 260 sheep.
260

---

## E.8 VALIDATION OF PRESERVED FUNCTIONAL HIERARCHY

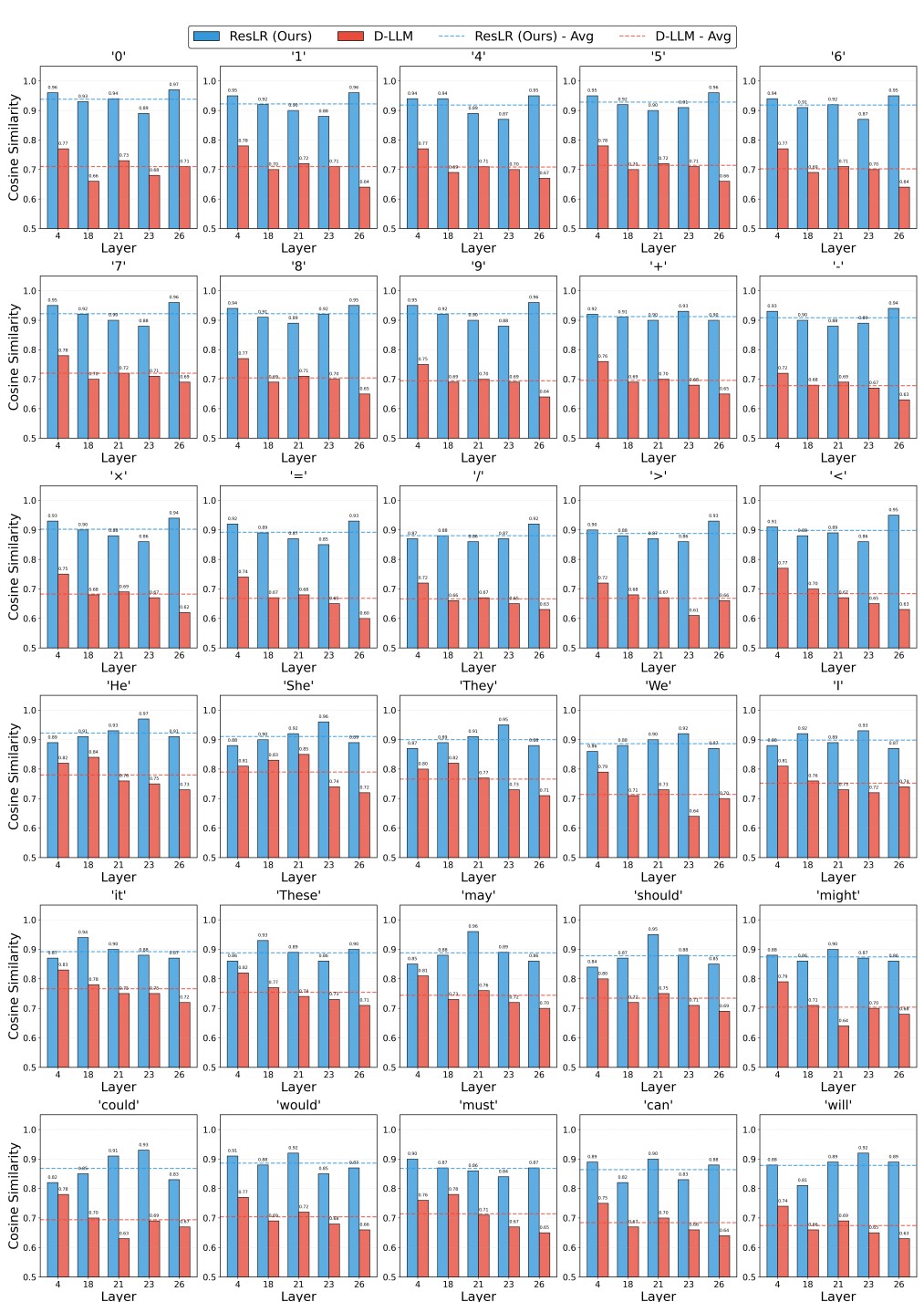

Figure 17: Validation of preserved functional hierarchy. Using the full-sized LLM as the baseline, we extracted features from various layer positions in both ResLR and D-LLM. Cosine similarity is calculated between these extracted features and those from the corresponding layers of the baseline model. The results demonstrate that our method achieves approximately 90% feature similarity with the full-sized model at identical positions. This represents an improvement of about 26% over D-LLM, directly supporting our claim that the proposed method preserves the functional hierarchy.

# F APPENDIX: LLM USAGE STATEMENT

**Use of LLMs:** Yes

- Purpose: Writing assistance and polishing. LLMs were used to refine language, improve clarity, and enhance readability without altering technical conclusions or empirical results.

All LLM-assisted content was reviewed and edited by the authors; key findings and data-driven conclusions do not rely on LLM-generated content.

