# OpenReview forum: "ResLR: Residual-Low-Rank Surrogates for Stable and Fast Context Adaptive Computing in Large Language Models"
_ICLR.cc/2026/Conference — Submitted to ICLR 2026_

### Official Review · Reviewer_f4Zp · 2025-10-19

**Soundness:** 2
**Presentation:** 2
**Contribution:** 2
**Rating:** 2
**Confidence:** 3

**Summary:**

This paper addresses routing instability in token-wise layer-skipping methods for efficient LLM inference. The authors propose ResLR surrogates, which is a low-rank approximation of multi-layer residuals trained via self-distillation—combined with block-wise routing that makes unified decisions for groups of layers. Experiments on LLaMA2-7B/13B show 48-52% FLOPs reduction with improved stability (53.7% lower routing variance) and performance gains over baselines.

**Strengths:**

1. The problem is well motivated. The paper identifies the routing instability issue with existing layer-skipping methods caused by violating functional hierarchy and per-layer routing volatility. The empirical evidence (Figure 2) effectively demonstrates this problem.

2. The experimental results are sound, as table one clearly shows empirical gains.

**Weaknesses:**

1. The self-distillation formulation Is Circular. In particular, equation defines the main loss term, where E_{i+1} is the "teacher" output. But the paper never clarifies if E_{i+1} is computed with or without the surrogate f participating in the forward pass. If the surrogate participates: The target E_{i+1} depends on f, creating a moving target problem. This invalidates the entire theoretical analysis, as lemma 1 assumes approximating a fixed residual \delta E, but here \delta E depends on the thing being trained. The bias-variance decomposition (Eq. 9) breaks down because \phi* is no longer well-defined. If the surrogate is frozen: This should be stated explicitly, and you need ablations showing training stability with/without this constraint.The self-distillation literature addresses these circularity issues with EMA teachers or momentum updates. Your formulation ignores this entirely, making the theoretical contribution questionable.

2. "Preserving Functional Hierarchy" Is not validated. The core claim is that ResLR preserves the model's functional hierarchy better than direct skipping (lines 79-91), but this is never directly tested. The evidence provided is indirect: higher mutual information just shows correlated decisions, not functional correctness; lower variance shows stability, not hierarchy preservation; better task performance has many possible explanations.

**Questions:**

The paper claims all baselines use "self-distillation joint training" but D-LLM's original paper doesn't. Did you retrain everything under your protocol? If so, are the comparisons fair to methods designed for different training regimes?

Also: the paper only compares layer-skipping methods. Speculative decoding achieves 2-3× speedups with zero accuracy loss. How does ResLR compare? Claiming "state-of-the-art" without comparing these is misleading.

---

> ### Author Response · Authors · 2025-11-13
> **Interim Response to Reviewer f4Zp: Clarification of Circular Argumentation**
>
> **Our implementation has already aligned with your point** that "$E_{i+1}$ should be computed without the surrogate $f$ participating in the forward pass." To illustrate this, we will cite a code snippet from our original submission ("code/llama/model_train.py" in Supplementary Material).
> The layer-wise distill loss is calculated in **L953-957**:
> ```python
> h, w, full_output, block_info = output
> #(...)
> layer_distill_loss = self.criterion_distill(h, full_output)
> ```
> The two input variables are derived from the values returned by the layer.
> **L843-845:**
> ```python
> if hasattr(self, 'distill_mode') and self.distill_mode:
>     mixed_output = w * full_output + (1 - w) * low_rank_output
>     return mixed_output, w, full_output, block_info
> ```
> Where the `full_output` is generated via the standard Transformer block:
>
> **L822-L823:**
> ```python
> h = x + self.attention(self.attention_norm(x), start_pos, freqs_cis, mask, eviction_mask)
> full_output = h + self.feed_forward(self.ffn_norm(h))
> ```
>
> And the `low_rank_output` is produced by the approximator:
>
> **L829:**
> ```python
> low_rank_output = self.multi_path_approximator(x, path_segments_per_token)
> ```
>
> Given that the weight `w` is the result of a sampling process `F.gumbel_softmax(x, dim=-1, tau=1, hard=True)` (L350), this implementation gives rise to two distinct cases:
> -  **When `w=1`:** The distillation loss becomes zero, and no action is taken.
> -  **When `w=0`:** The Mean Squared Error (MSE) between `full_output` and `low_rank_output` is calculated. The computation of `full_output` and `low_rank_output` follows two independent paths.
>
> The key aspect of the implementation is that `w`, as the output of the Gumbel-Softmax function with `hard=True`, produces discrete path selections in the forward pass. Importantly, the `full_output` is computed independently by the complete Transformer block, while the `low_rank output` is computed independently by the approximator. The computational processes for these two outputs do not depend on each other. The weight `w` used to mix the outputs is a binary value, meaning when one channel is activated, the other is completely shut off. Therefore, when the distillation loss is calculated, the `full output` serves as a fixed teacher signal, and no circular dependency exists.
>
> Additionally, we acknowledge that, in light of our code implementation, Equations (6) and (12) in their current form may be misleading. In Equation (6), the term $E_{i+1}$, which implies a positional dependency, should be replaced with $E_{\text{Transformer}}$. This change would better reflect that the teacher signal is derived from a separate, parallel path. Similarly, the description for Equation (12) should be updated to emphasize that Gumbel-Softmax sampling is implemented to ensure $r_{s, e}$ is a binary value, rather than a continuous weighting coefficient.
>
> Please consider this an interim response. We will address the remainder of your concerns and upload a revised manuscript within the next 1-2 days.

---

> ### Author Response · Authors · 2025-11-14
> **Official Response to Reviewer f4Zp: Other Technical Concerns**
>
> **On the concern regarding functional hierarchy preservation**: In the revised manuscript, We make the claim verifiable by (i) clarifing the operational definition and (ii) adding new discuss section along with visualization.
> - **Operational definition** In the updated introduction, we clarify that the concept of a large model's "functional hierarchy" is not a novel concept introduced or summarized by us. It has been operationally defined in prior work, such as D-LLM, where layers at different positions within the model are associated with specific semantic features. Consequently, we establish the preservation of the pre-trained model's functional hierarchy as a key design objective.
>
> - **New analyses** We add a paragraph in experiment section and an appendix section (Appendix E.8) that illustrate how ResLR realizes functional hierarchy preservation. Among these findings, the dataset-level mean layer activation rates (as shown in Figure 9) reveal that D-LLM exhibits more extreme activation patterns, with most layers having rates close to either 0 or 1. This indicates that D-LLM learns a new functional hierarchy rather than preserving the depth-to-function mapping of the pre-trained LLM. In contrast, the layer activation rates for ResLR are considerably smoother.
>
> Furthermore, we selected several representative tokens for a more direct comparison, as shown in Figure 17. Using the full-sized LLM as the baseline, we extracted features from various layer positions in both ResLR and D-LLM via probing. We then calculated the cosine similarity between these extracted features and those from the corresponding layers of the baseline model. The results demonstrate that our method achieves approximately 90% feature similarity with the full-sized model at identical positions. This represents an improvement of about 26% over D-LLM, directly supporting our claim that the proposed method preserves the functional hierarchy.
>
> **Comparison fairness**: Self-distillation is a distinctive step within our proposed method, applied exclusively to the surrogate model. Its sole purpose is to constrain the behavior of the surrogate path to align with the main path; it is not intended to directly enhance the model's task performance. In the main results table, we only compare the methods that include LoRA fine-tuning. Meanwhile, in our revised implementation details, we clarify that the same LoRA rank was used for fine-tuning our model as in our reproduction of D-LLM. Furthermore, our reproduced D-LLM still outperforms the other methods, which is consistent with the results reported in the original D-LLM paper. This ensures a fair comparison under identical settings. We   have also changed the confusing expression "**we fine-tuned the models...**" to “**we fine-tuned our method...**”, we apologize for the misunderstanding caused.
>
> **Orthogonality to speculative decoding**: We acknowledge the reviewer's insightful comparison between dynamic inference and speculative decoding as two distinct acceleration strategies. However, we would like to point out that these two approaches are, in principle, orthogonal.
>
> State-of-the-art speculative decoding methods, such as Medusa[1] and the EAGLE series[2-4], enhance decoding efficiency by building upon pre-trained neural network backbones and adding parallel decoding heads. In contrast, ResLR and the dynamic inference paradigm it belongs to, focus on improving the computational efficiency of the backbone itself. The orthogonality of the two acceleration methods lies in the following:
>
> - For methods that utilize an LLM backbone, such as Medusa and EAGLE, the core idea is to employ parallel decoding heads to achieve acceleration. In this context, ResLR serves to reduce the computational resource demands of the LLM backbone.
>
> - For the broader category of draft model-based methods, ResLR can be concurrently applied to both the large language model and the draft model, provided that both architectures are Transformer-based.
>
> [1] T. Cai, et al., "Medusa: Simple LLM Inference Acceleration Framework with Multiple Decoding Heads," in Forty-first International Conference on Machine Learning (ICML 2024), Vienna, Austria, 2024. [Online]. Available: https://openreview.net/forum?id=PEpbUobfJv
>
> [2] Y. Li, F. Wei, C. Zhang, and H. Zhang, "EAGLE: Speculative Sampling Requires Rethinking Feature Uncertainty," in Forty-first International Conference on Machine Learning (ICML 2024), Vienna, Austria, 2024. [Online]. Available: https://openreview.net/forum?id=1NdN7eXyb4
>
> [3] Y. Li, F. Wei, C. Zhang, and H. Zhang, "EAGLE-2: Faster Inference of Language Models with Dynamic Draft Trees," in Proc. 2024 Conf. on Empirical Methods in Natural Language Processing (EMNLP 2024), Miami, FL, USA, 2024, pp. 7421–7432. DOI: 10.18653/v1/2024.emnlp-main.422.
>
> [4] Y. Li, F. Wei, C. Zhang, and H. Zhang, "EAGLE-3: Scaling up Inference Acceleration of Large Language Models via Training-Time Test," arXiv preprint arXiv:2503.01840, 2025.

---

> ### Author Response · Authors · 2025-11-24
> **Official Comment by Submission6409 Authors**
>
> Dear Reviewer f4Zp,
>
> Thank you for your rigorous review.
>
> We are writing to kindly check if our response has helped clarify the technical concerns. Specifically, we have:
> 1.  Provided **code snippets** to demonstrate the **independence of the teacher signal**, addressing the concern regarding circular argumentation.
> 2.  Included **new probing analyses** to directly validate the preservation of functional hierarchy.
> 3.  Discussed the **orthogonality** between our method and speculative decoding.
>
> We hope these clarifications fully resolve your doubts, and we remain open to any further discussion.
>
> Best regards,
>
> Submission 6409 Authors

---

### Official Review · Reviewer_XiPs · 2025-10-28

**Soundness:** 3
**Presentation:** 2
**Contribution:** 2
**Rating:** 4
**Confidence:** 2

**Summary:**

This paper addresses LLM inference costs through dynamic layer skipping. The authors identify routing instability in existing methods and propose ResLR with two components: (1) low-rank surrogates to approximate skipped layers, motivated by SVD analysis showing 90% information retention in 50 components, and (2) block-wise routing that groups adjacent layers for unified decisions. Results show 48-52% FLOPs reduction with 1.9× speedup and improved stability vs D-LLM.

**Strengths:**

1. Well-motivated problem. Figure 2 quantifies routing instability with mutual information and variance metrics, providing clear motivation.

2. Reasonable design choices. SVD analysis supports the low-rank assumption. Block-wise routing addresses both efficiency and stability in a straightforward way.

3. Consistent improvements across 9 benchmarks with both FLOPs reduction and wall-time speedup.

**Weaknesses:**

1. There are some reproducibility concerns, as several essential implementation details are not fully specified:
- Specific rank r values and their selection criteria
- Gating network architecture and training procedure
- Loss balancing between distillation and task objectives
- ResLR insertion strategy (which layers, all or selective?)

With these details clarified, it would be easier to reproduce the method.

2. The paper lacks explanation for why inter-layer residuals have low-rank structure. More critically, the claim that ResLR "preserves functional hierarchy" is not rigorously justified. How does low-rank projection preserve hierarchy differently than direct skipping? Without probing experiments or formal analysis, this core claim remains unsubstantiated.

3. Testing only 7B and 13B is insufficient for "excellent scalability" claims. No evaluation on:
- Models larger than 13B (30B, 70B)
- Long-context scenarios (8k+ tokens)
- Different architectures beyond LLaMA

The generalization remains questionable.

**Questions:**

1. What specific rank r do you use? How is it selected?The paper mentions r << d but never specifies actual values. Is it the same across all layers? This is fundamental for reproducibility.
2. How exactly does ResLR preserve functional hierarchy? This is your core claim, but the mechanism is unclear. Can you provide probing experiments or formal analysis showing that low-rank projection preserves hierarchy better than direct skipping?
3. Does the low-rank structure hold across different models and tasks?
4. Why is block-wise routing necessary if ResLR already addresses the hierarchy problem?

---

> ### Author Response · Authors · 2025-11-14
> **Official Response to Reviewer XiPs: Reproducibility Concerns**
>
> We acknowledge that due to page limitations, the description in the main paper was concise, with most details deferred to the appendix, requiring frequent jumps \& navigation. In the revised version, we have enhanced the "reproducibility statement" section by incorporating centralized hyperlinks to improve reader readability. Meanwhile, we would like to address your concerns point-to-point:
>
> - **Specific rank $r$ values and their selection criteria**: rank $r$ value is set to a value of 256. This value was determined jointly by the theoretical formulation presented in Equation (11) of the paper (as noted by Reviewer gncQ in Strength 3) and an empirical analysis of hyperparameter sensitivity, detailed in Appendix D.
>
> - **Gating network architecture and training procedure**: We acknowledge the need for greater clarity regarding the router's architecture. Although the router's structure was illustrated in Figure 3 and its specific dimensions were detailed in Appendix C of the initial submission, we agree that defining its structure at its first mention (Section 3.2) is essential—as we have now highlighted in the corresponding section of the revised manuscript. Furthermore, we wish to clarify that the router is trained concurrently with the LoRA fine-tuning process. It is optimized jointly through the task loss and the Activation-Rate Regularization (as defined in Equation 13). This simultaneous optimization is crucial for realizing one of our key contributions: a one-stage fine-tuning framework.
>
> - **Loss balancing between distillation and task objectives**: As demonstrated in Equation 14 in the main text, the total optimization objective is a weighted sum of three loss components. In the initial submission, we stated that $λ_{distill}$ is set to 1.0 and that "$λ_{active}=10.0$ for all tasks to balance the loss terms." In the revised version, we clarify the rationale for setting $λ_{active}$ to 10.0: the main task loss typically converges to a similar order of magnitude during the optimization process. A hyperparameter sensitivity analysis for these parameters is also provided in Appendix D.
>
> - **ResLR insertion strategy**: The insertion of ResLR is performed on a block-wise basis. As detailed in Appendix B, a single block initializes $β(β+1)/2$ ResLR surrogates to cover all possible sub-paths within it. The activation and deactivation of these surrogates are governed by a unified block-level router.

---

> ### Author Response · Authors · 2025-11-14
> **Official Response to Reviewer XiPs: Technical Concerns**
>
> **Response on functional hierarchy preservation concerns**:In the revised manuscript, We make the claim verifiable by (i) clarifing the operational definition and (ii) adding new discuss section along with visualization.
> - **Operational definition** In the updated introduction, we clarify that the concept of a large model's "functional hierarchy" is not a novel concept introduced or summarized by us. It has been operationally defined in prior work, such as D-LLM, where layers at different positions within the model are associated with specific semantic features. Consequently, we establish the preservation of the pre-trained model's functional hierarchy as a key design objective.
>
> - **New analyses** We add a paragraph in experiment section and an appendix section (Appendix E.8) that illustrate how ResLR realizes functional hierarchy preservation. Among these findings, the dataset-level mean layer activation rates (as shown in Figure 9) reveal that D-LLM exhibits more extreme activation patterns, with most layers having rates close to either 0 or 1. This indicates that D-LLM learns a new functional hierarchy rather than preserving the depth-to-function mapping of the pre-trained LLM. In contrast, the layer activation rates for ResLR are considerably smoother.
>
> Furthermore, we selected several representative tokens for a more direct comparison, as shown in Figure 17. Using the full-sized LLM as the baseline, we extracted features from various layer positions in both ResLR and D-LLM via probing. We then calculated the cosine similarity between these extracted features and those from the corresponding layers of the baseline model. The results demonstrate that our method achieves approximately 90% feature similarity with the full-sized model at identical positions. This represents an improvement of about 26% over D-LLM, directly supporting our claim that the proposed method preserves the functional hierarchy.
>
> **Response for generalization concerns**: As also suggested by Reviewer gncQ, we conducted additional experiments on LLaMA3-8B and Qwen3-8B following our initial submission. The results have been updated in Table 1 of the revised manuscript.
>
> We acknowledge that, due to hardware constraints, we were unable to perform experiments on models exceeding 13B parameters. However, we would like to point out that the representative concurrent works compared in Table 1 also conduct their validation solely on models of 8B or smaller. In the revised version, we have **refined the imprecise claim** of "excellent scalability" to "**This design is intended to preserve the functional hierarchy of the original model, thus exhibiting scalability and generalization ability.**"
>
> **Response for questions**:
> - **Does the low-rank structure hold across different models and tasks**: Yes, as validated in Appendix E.1.
> - **Why is block-wise routing necessary if ResLR already addresses the hierarchy problem**: The block router and ResLR originate from the same observation of layer-wise feature similarity. As an engineering optimization, the block router helps to stabilize the training path for ResLR. Ablation studies demonstrate that both components not only contribute independently but also work synergistically to achieve simultaneous improvements in both FLOPs and performance.

---

> ### Author Response · Authors · 2025-11-24
> **Official Comment by Submission6409 Authors**
>
> Dear Reviewer XIPs,
>
> Thank you for your detailed feedback, particularly regarding reproducibility and the theoretical validation.
>
> We are writing to ensure that our response has effectively addressed your concerns. specifically, we have provided:
> 1.  **Clarifications on implementation details** (e.g., specific rank selection $r=256$, gating network structure) to ensure reproducibility.
> 2.  **New probing analyses and visualizations** to empirically substantiate the preservation of "functional hierarchy."
> 3.  **Additional generalization results** on LLaMA3-8B and Qwen3-8B.
>
> We are eager to know if these updates have resolved your doubts and stand ready to provide further clarification if needed.
>
> Best regards,
>
> Submission6409 Authors

---

### Official Review · Reviewer_gncQ · 2025-10-31

**Soundness:** 3
**Presentation:** 3
**Contribution:** 3
**Rating:** 6
**Confidence:** 4

**Summary:**

This paper addresses a critical bottleneck in deploying Large Language Models (LLMs): high inference cost due to every token passing through all layers of the Transformer stack. While recent work has explored layer skipping to reduce this cost, such methods often suffer from unstable routing decisions and degrade the model’s internal functional structure, leading to volatile accuracy and inefficient fine-tuning.

To solve this, the authors propose ResLR, a unified acceleration framework that improves both efficiency and stability in dynamic inference. It combines two key innovations:
1。 ResLR surrogates that replaces skipped Transformer layers with a learned low-rank operator, preserving the model’s functional hierarchy.
2. Block-Wise Multi-Path Routing: Instead of deciding skip decisions per layer, the model groups layers into blocks and makes a single routing decision per block, reducing overhead and stabilizing the computation.

This framework is plug-and-play with standard LoRA fine-tuning and demonstrates state-of-the-art trade-offs on multiple reasoning and language benchmarks, reducing FLOPs by over 50%, increasing inference speed by 1.9×, and improving output stability over previous dynamic approaches.

**Strengths:**

1. The paper identifies two causes of instability in prior layer-skipping methods—(1) disrupting the model’s learned depth hierarchy, and (2) noisy, per-layer routing without inter-layer coordination.
2. The proposed ResLR surrogate is proposed to learn to approximate the combined residual transformation of skipped layers using a low-rank structure, trained via self-distillation, and achieve good empirical results.
3. The paper provides a clear bias-variance decomposition for the surrogate’s approximation error and justifies its rank selection.
4. The method can be integrated seamlessly into existing LoRA fine-tuning pipelines.

**Weaknesses:**

1. Unlike standard LoRA or other low-rank fine-tuning methods, the proposed ResLR surrogate operates as an external bypass module rather than a low-rank correction to existing Transformer weights. As a result, the trained ResLR components cannot be merged back into the original model weights, making deployment more complex and limiting compatibility with existing model-serving optimizations that rely on weight merging (e.g., static inference graph export).
2. Because the dynamic router determines which layers to execute per token, the method requires custom inference logic and is not directly compatible with existing static inference engines, possibly complicating real-world deployment.
3. While tested on LLaMA2, the method’s performance and compatibility with other LLM families (e.g., Qwen3, Llama3.x) is not assessed.

**Questions:**

While block-wise routing amortizes decision cost, inference still requires executing token-level gating logic. In high-throughput or multi-token batch settings, how significant is this routing overhead, especially as block size or model size increases?

---

> ### Author Response · Authors · 2025-11-14
> **Official Response to Reviewer gncQ**
>
> 1.**Response to W1, W2 and Q1** We find your concerns regarding the adaptability of dynamic model structures to current inference frameworks to be insightful. We also acknowledge that the surrogate paths cannot be merged back into the original model. However, this bears a strong resemblance to the recently popularized Mixture-of-Experts (MoE) architecture. Specifically, the original transformer path and the surrogate path can be compiled together as static subgraphs, akin to an MoE model with an expert count of K=2, where the two "experts" represent the original and surrogate paths, respectively. At runtime, a binary mask is used to scatter the active tokens to the chosen subgraph (either the original or the surrogate) and then gather the corresponding outputs.This approach preserves the ability for static graph export while enabling token-level dynamic routing. Consequently, optimization strategies for this execution model would be shared with those for MoE architectures[1, 2]. We will include this as a direction for future work in the concluding discussion of the paper.
>
> In response to your inquiry regarding the impact of block size on the additional overhead of decision-making, we present the following table. It illustrates the differences in speedup for the ResLR method across various block sizes, with a block size sweep from 1 to 16, under an relative FLOPS rate of ~0.5. It is important to note that the actual wall-clock time is more significantly influenced by FLOPS, which may lead to fluctuations in execution time. However, the linearity of the speedup demonstrates a consistent upward trend relative to the block size.
>
> | Block Size | FLOPs | Wall time (ms) | Linearity |
> |---|-------|-----------|-----------|
> | β=1 | 0.53 | 23.876 | 85.2% |
> | β=2 | 0.54 | 22.703 | 91.3% |
> | β=4 | 0.51 | 20.704 | 94.6% |
> | β=8 | 0.51 | 20.313 | 96.4% |
> | β=16 | 0.53 | 20.879 | 97.4% |
>
> 2.**Response to W3** We fully agree with your request for evidence regarding compatibility with other LLM families. After our initial submission, we commenced experiments on LLaMA3-8B and Qwen3-8B, achieving consistent state-of-the-art results. For detailed outcomes, please refer to the revised Table 1 in our updated manuscript.
> Furthermore, we would like to highlight that the generalization ability of the ResLR surrogate is fundamentally determined by its theoretical derivation—specifically, by isolating the existing pass-through path within the Transformer layer structure and approximating its residual component. This foundational principle is independent of the specific implementation details of the Transformer architecture across different model families.
>
> [1] O. Balmau, A.-M. Kermarrec, R. Pires, A. L. E. Santo, M. de Vos, and M. Vujasinovic, "Accelerating MoE Model Inference with Expert Sharding," in Proceedings of the 5th Workshop on Machine Learning and Systems, New York, NY, USA: Association for Computing Machinery, 2025, pp. 192–199. doi: 10.1145/3721146.3721940.
>
> [2]Z. Jin et al., "BigMac: a communication-efficient mixture-of-experts model structure for fast training and inference," in Proceedings of the Thirty-Ninth AAAI Conference on Artificial Intelligence, 2025, Art. no. 1972, pp. 1-10. doi: 10.1609/aaai.v39i17.33945.

---

> ### Author Response · Authors · 2025-11-24
> **Official Comment by Submission6409 Authors**
>
> Dear Reviewer gncQ,
>
> Thank you again for your time and the constructive feedback.
>
> We are writing to kindly ensure that our response—**specifically the discussion on MoE-like inference compatibility (addressing W1 & W2) and the new experiments on LLaMA3 & Qwen3 (addressing W3)**—has sufficiently resolved your concerns.
>
> **We remain available to answer any further questions or provide additional details to assist your final assessment**.
>
> Best regards,
>
> Submission6409 Authors

---

### Author Response · Authors · 2025-11-14
**Thanks for the Reviews: Revision Summary**

We thank the reviewers for their insightful and constructive feedback. This official comment aims to summarize the revisions made to the manuscript.

### Revision #1 summary

- **New results**: Llama3-8B and Qwen3-8B experiments, appended after the main results table 1.

- **Evidence for functional hierarchy preservation**: We have augmented the manuscript with additional experimental sections and an Appendix E.8. Furthermore, in the introduction, we have clarified the operational definition of "functional hierarchy preservation" to substantiate the central claim of our paper. Quantitative analysis of our experiments reveals that our approach preserves approximately 90% of the functionalities of the pre-trained LLM. For a more detailed discussion, please refer to our individual responses to Reviewer f4Zp or XiPs.

- **Presentation fixes**: Clarifying definitions, standardizing notation, and elaborating on details.

- **Enhanced reproducibility checklist**: We acknowledge that due to page limitations, the description in the main paper was concise, with most details deferred to the appendix, requiring frequent jumps\&navigation. In the revised version, we have enhanced the "reproducibility statement" section by incorporating centralized hyperlinks to improve reader readability.

The revised sections are highlighted in yellow.

---

### Author Response · Authors · 2025-11-29
**Appreciation to the AC and Reviewers: Summary of Rebuttal Discussions**

We sincerely appreciate the AC's dedication and fully recognize the additional workload imposed by the discussion rollback following the information leakage. **To assist the AC in navigating this situation and to facilitate an efficient decision-making process, we provide this summary, which we guarantee is strictly derived from the preserved records of the actual discussion.**

### Strengths \& Contributions
Here, we summarize **the strengths acknowledged by the majority of reviewers (≥2 reviewers)** to demonstrate that our work delivers a full-cycle contribution, encompassing problem diagnosis, solution, and empirical results.

1. **Well-motivated problem** (all reviewers): Our work identifies the routing instability issue with existing layer-skipping methods caused by violating functional hierarchy and per-layer routing volatility.

2. **Reasonable design choices** (reviewer gncQ \& XiPs): The ResLR surrogate is designed to approximate the low-rank residual flow between LLM layers, where the low-rank nature is validated by SVD results and the structural design is theoretically derived.

3. **Good empirical results** (all reviewers): We achieved consistent improvements across 9 benchmarks with both FLOPs reduction and wall-time speedup on both 7B and 13B LLaMA-2 model. (In the revised version, we conducted additional experiments verifying that the aforementioned improvements also hold for LLaMA3 and Qwen3.)

### Reviewer's Concerns \& Our Response

Here, we summarize **our point-by-point responses to all raised Weaknesses and Questions**. These concerns are organized by priority from the authors' perspective.

1. **Circular Argumentation Issue** (see reviewer f4Zp): We provide evidence (specifically, the code implementation in the supplementary material) demonstrating that **this concern stems from a misunderstanding of the phrasing in the original manuscript. We have identified the potential cause of this ambiguity and clarified it in the revised version.**

2. **Concern on Functional Hierarchy Preservation** (see reviewer f4Zp \& XiPs): In the revised version, we first **clarified the operational definition** of Functional Hierarchy Preservation in the introduction. Additionally, we **included a new experimental subsection** (Illustration of Functional Hierarchy Preservation) and **a new appendix subsection** (Appendix E.8 Validation of Preserved Functional Hierarchy) to provide supplementary evidence.

3. **Generalization Concern** (see reviewer XiPs \& gncQ): In the revised version, we **added results for LLaMA3 and Qwen3** across the original 9 benchmarks under identical experimental settings to substantiate our method's generalizability. We further highlight that this generalizability stems from the decomposition of the universal Transformer computational path (Eq.3) and the design philosophy of preserving original model's functionality.

4. **Reproducibility Concerns** (see reviewer XiPs): We provide evidence that the original submission **already included all relevant reproducibility details**, covering the code, model architecture, and hyperparameter sensitivity analysis. In the revised version, we have enhanced the reproducibility statement to provide a unified index for these details.

5. **Comparison Fairness** (see reviewer f4Zp): We clarify that **all compared methods are dynamic inference approaches that involve fine-tuning**. When using D-LLM as a baseline, we adopted the identical experimental settings as those used for our proposed method. Furthermore, our reproduced performance for D-LLM is **consistent with the officially reported results**.

6. **Adaptability with Existing Acceleration Frameworks** (see reviewer gncQ): We demonstrate compatibility with existing acceleration frameworks by **highlighting the similarity** between our proposed routing-controlled "compute-approximate" dual-path structure and a **Mixture-of-Experts (MoE) architecture with $K=2$**. Furthermore, we point out that efficiently scattering embeddings and gathering results during batched inference is a key focus for future hardware optimization. As this challenge mirrors those found in MoE structures, we have included a discussion on this in the Conclusion of the revised version.

7. **Discussion on Questions**: We further address the minor issues raised by the reviewers, including the impact of block size on acceleration (Reviewer gncQ), specific methodological design choices (Reviewer XiPs), and the orthogonality of our proposed method with other acceleration frameworks (specifically, Speculative Decoding, Reviewer f4Zp)."

---

### Meta-Review · Area_Chair_iBBb · 2026-01-07

**Summary:**

The paper presents a new approach, ResLR, that aims to improve the efficiency and stability of dynamic inference with LLMs. The method replaces skipped layers with a learned low-rank operator, and making skip decisions not per layer, but per block, for further efficiency.

 The motivation of the paper is clear, reviewers deem the design choices sensible, and the method shows strong and largely comprehensive experimental results, both in terms of FLOPS and wall-time speedup on 9 benchmarks. Furthermore, the authors have addressed some major concerns of the reviewers (e.g., generalization to other models) during the rebuttal process.

 The reviewer had concerns about compatibility with existing inference framework or limited comparison with other efficiency optimization methods.  While some efficiency optimization methods (e.g., speculative decoding, as proposed by reviewer) can be deemed orthogonal to this approach as authors claim, it is also not clear how to integrate such orthogonal approaches together. It'd be helpful to discuss other efficiency optimization methods (of different nature) a bit more comprehensively in related work.

The reviewers also raised concerns about the clarity of the algorithm description and claims about functional hierarchy preservation. The figures are helpful but a clear pseudocode / algorithm block would help with the readability and clarity of the paper.  Overall, the paper presents a promising idea with convincing experimental results, but hard to recommend in its current form for above weaknesses.

**Reviewer Concerns:**

The reviewer gncQ had concerns about compatibility of existing inference framework. The authors largely acknowledged this issue.  The reviewer XiPs raised concerns about generalization, which authors partially addressed (Different model family), but not entirely (no larger model or long context scenarios). The reviewer also raised some clarity issue with the paper, few key details missing from the paper.

**Reviewer Scores:**

The reviewer qncQ would have increased score, as they showed results on other LLM families, showing positive results, as reviewer wanted to see.

---

### Decision · Program_Chairs · 2026-01-26

Reject